# TO GROK OR NOT TO GROK: DISENTANGLING GENERALIZATION AND MEMORIZATION ON CORRUPTED ALGORITHMIC DATASETS

**Darshil Doshi** [a, b, †]     **Aritra Das** [b, *]     **Tianyu He** [a, b, *]     **Andrey Gromov** [c, a, b]

{ddoshi, aritrad, tianyuh}@umd.edu         gromovand@meta.com

## ABSTRACT

Robust generalization is a major challenge in deep learning, particularly when the number of trainable parameters is very large. In general, it is very difficult to know if the network has memorized a particular set of examples or understood the underlying rule (or both). Motivated by this challenge, we study an interpretable model where generalizing representations are understood analytically, and are easily distinguishable from the memorizing ones. Namely, we consider multi-layer perceptron (MLP) and Transformer architectures trained on modular arithmetic tasks, where ($\xi \cdot 100\%$) of labels are corrupted (*i.e.* some results of the modular operations in the training set are incorrect). We show that (i) it is possible for the network to memorize the corrupted labels *and* achieve $100\%$ generalization at the same time; (ii) the memorizing neurons can be identified and pruned, lowering the accuracy on corrupted data and improving the accuracy on uncorrupted data; (iii) regularization methods such as weight decay, dropout and BatchNorm force the network to ignore the corrupted data during optimization, and achieve $100\%$ accuracy on the uncorrupted dataset; and (iv) the effect of these regularization methods is ("mechanistically") interpretable: weight decay and dropout force all the neurons to learn generalizing representations, while BatchNorm de-amplifies the output of memorizing neurons and amplifies the output of the generalizing ones. Finally, we show that in the presence of regularization, the training dynamics involves two consecutive stages: first, the network undergoes *grokking* dynamics reaching high train *and* test accuracy; second, it unlearns the memorizing representations, where the train accuracy suddenly jumps from $100\%$ to $100(1-\xi)\%$.[1]

## 1 INTRODUCTION

The astounding progress of deep learning in the last decade has been facilitated by massive, high-quality datasets. Annotated real-world datasets inevitably contain noisy labels, due to biases of annotation schemes (Paolacci et al., 2010; Cothey, 2004) or inherent ambiguity (Beyer et al., 2020). A key challenge in training large models is to prevent overfitting the noisy data and attain robust generalization performance. On the other hand, in large models, it is possible for memorization and generalization to coexist (Zhang et al., 2017; 2021). By and large, the tussle between memorization and generalization, especially in the presence of label corruption, remains poorly understood.

In generative language models the problem of memorization is even more nuanced. On the one hand, some factual knowledge is critical for the language models to produce accurate information. On the other hand, verbatim memorization of the training data is generally unwanted due to privacy concerns (Carlini et al., 2021). The full scope of the conditions that affect memorization in large

---

[a]Condensed Matter Theory Center, University of Maryland, College Park
[b]Department of Physics, University of Maryland, College Park
[c]Meta AI
[†]Corresponding author
[*]These authors contributed equally

[1]Code to reproduce our results is available at https://github.com/d-doshi/Grokking.git

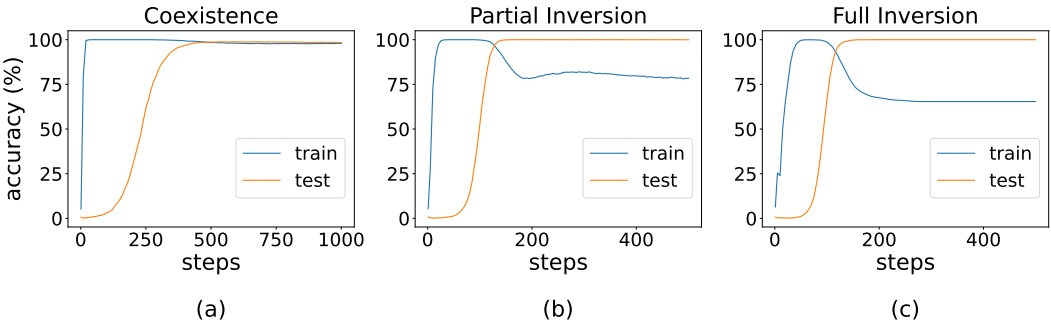

Figure 1: Training curves in various phases. All plots are made for networks trained with data-fraction $\alpha = 0.5$ and corruption-fraction $\xi = 0.35$. (a) No regularization: *Coexistence* of generalization and memorization – both train and test accuracies are high. (b)(c) Adding weight decay: The network generalizes on the test data but does not memorize the corrupted training data, resulting in a negative generalization gap! Remarkably, the network predicts the "true" labels for the corrupted examples. We term these phases *Partial Inversion* and *Full Inversion*, based on the degree of memorization of corrupted data. ("inversion" refers to test accuracy being higher than train accuracy.)

language models (LLMs) is still under investigation (Carlini et al., 2022; Tirumala et al., 2022). In particular, larger models as well as repeated data (Hernandez et al., 2022) favor memorization.

In this work, we take an approach to disentangling generalization and memorization motivated by theoretical physics. Namely, we focus on simple, analytically tractable, ("mechanistically") interpretable models trained on algorithmic datasets. More concretely, we study two-layer Multilayer Perceptrons (MLP) trained on modular arithmetic tasks. In this case, the network is trained to learn a rule, such as $z = (m + n)\%p \equiv m + n \bmod p$ from a number of examples. These tasks are phrased as classification problems. We then force the network to memorize $\xi \cdot 100\%$ of the examples by corrupting the labels, *i.e.* injecting data points where $(m + n)\%p$ is mapped to $z' \neq z$. In this setting, the generalizing representations are understood qualitatively Nanda et al. (2023); Liu et al. (2022) as well as quantitatively, thanks to the availability of an analytic solution for the representations described in Gromov (2023). We empirically show that sufficiently large networks can memorize the corrupted examples while achieving nearly perfect generalization on the test set. The neurons responsible for memorization can be identified using inverse participation ratio (IPR) and pruned away, leading to perfect accuracy on the un-corrupted dataset. We further show that regularization dramatically affects the network's ability to memorize the corrupted examples. Weight decay, Dropout and BatchNorm all prevent memorization, but in different ways: Weight decay and Dropout eliminate the memorizing neurons by converting them into generalizing ones, while Batch-Norm de-amplifies the signal coming from the memorizing neurons without eliminating them.

## 1.1 Preliminaries

**Grokking modular arithmetic.** We consider the modular addition task $(m + n)\%p$. It can be learned by a two-layer MLP. Explicitly, the network function takes form

$$\boldsymbol{f}(m, n) = \boldsymbol{W}\phi\left(\boldsymbol{W}_{in}(\boldsymbol{e}_m \oplus \boldsymbol{e}_n)\right) = \boldsymbol{W}\phi\left(\boldsymbol{U}\,\boldsymbol{e}_m + \boldsymbol{V}\,\boldsymbol{e}_n\right). \quad (1)$$

Here, $\boldsymbol{e}_m, \boldsymbol{e}_n \in \mathbb{R}^p$ are `one_hot` encoded numbers $m, n$. "$\oplus$" denotes concatenation of vectors $(\boldsymbol{e}_m \oplus \boldsymbol{e}_n \in \mathbb{R}^{2p})$. $\boldsymbol{W}_{in} \in \mathbb{R}^{N \times 2p}$ and $\boldsymbol{W} \in \mathbb{R}^{p \times N}$ are the first and second layer weight matrices, respectively. $\phi$ is the activation function. $\boldsymbol{W}_{in}$ is decomposed into two $N \times P$ blocks: $\boldsymbol{U}, \boldsymbol{V} \in \mathbb{R}^{N \times p}$. $\boldsymbol{U}, \boldsymbol{V}$ serve as embedding vectors for $m, n$ respectively. $\boldsymbol{f}(m, n) \in \mathbb{R}^p$ is the network-output on one example pair $(m, n)$. The targets are `one_hot` encoded answers $\boldsymbol{e}_{(m+n)\%p}$.

The dataset consists of $p^2$ examples pairs; from which we randomly pick $\alpha p^2$ examples for training[2], and use the other $(1 - \alpha)p^2$ examples as a test set. We will refer to $\alpha$ as the *data fraction*.

This setup exhibits grokking: delayed and sudden occurrence of generalization, long after memorization (Power et al., 2022; Nanda et al., 2023; Liu et al., 2022; Gromov, 2023).

---

[2]Throughout the text, networks are trained with Full-batch AdamW and MSE loss, unless otherwise stated.

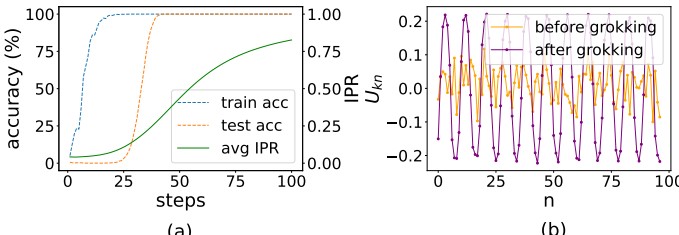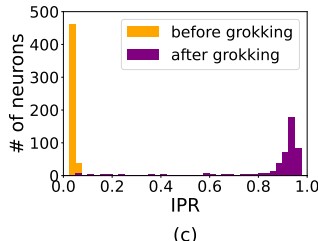

Figure 2: Grokking the modular arithmetic task over $\mathbb{Z}_{97}$ with 2-layer MLP, trained with AdamW. (a) Sharp transition in test accuracy, long after overfitting. $\overline{\text{IPR}} := \mathbb{E}_k\left[\text{IPR}_k\right]$ monotonically increases over time, indicating periodic representations. (b) Example row vector $(U_{k\cdot})$ before and after grokking. Generalization is achieved through periodic weights (equation 2). (c) Histogram of per-neuron IPRs before and after grokking – the distribution shifts to high IPR.

**Periodic weights.** Choosing quadratic activation function $\phi(x) = x^2$ makes the problem analytically solvable (Gromov, 2023). Upon grokking, the trained network weights are qualitatively similar to the analytical solution presented in Gromov (2023).[3] The following analytical expression for the network weights leads to $100\%$ generalization (for sufficiently large width)

$$U_{ki} = \left(\frac{2}{N}\right)^{-\frac{1}{3}} \cos\left(\frac{2\pi}{p}i\sigma(k) + \phi_k^{(u)}\right), \qquad V_{kj} = \left(\frac{2}{N}\right)^{-\frac{1}{3}} \cos\left(\frac{2\pi}{p}j\sigma(k) + \phi_k^{(v)}\right),$$

$$W_{qk} = \left(\frac{2}{N}\right)^{-\frac{1}{3}} \cos\left(-\frac{2\pi}{p}q\sigma(k) - \phi_k^{(u)} - \phi_k^{(v)}\right),$$

$$(2)$$

where $\sigma(k)$ denotes a random permutation of $k$ in $S_N$ – reflecting the permutation symmetry of the hidden neurons. The phase $\phi_k^{(u)}$ and $\phi_k^{(v)}$ are uniformly i.i.d. sampled between $(-\pi, \pi]$. We refer the reader to Appendix D for further details about the analytical solution.

**Inverse participation ratio.** To characterize the generalizing representations quantitatively, we utilize a quantity familiar from the physics of localization: the *inverse participation ratio (IPR)*. Its role is to detect periodicity in the weight matrix.

Let $U_{k\cdot}, V_{k\cdot}, (W_{k\cdot})$ denote the $k^{th}$ row(column) vectors of the weights $U, V(W)$. Consider the discrete Fourier transforms of these vectors, denoted by $\widetilde{U}_{k\cdot}, \widetilde{V}_{k\cdot}, (\widetilde{W}_{k\cdot})$. The Fourier decompositions of the periodic rows(columns) of these weights are highly localized. We leverage this to quantify the similarity of trained weights to the analytic solution (equation 2). To that end, we define the *Inverse Participation Ratio (IPR)* (Girvin & Yang, 2019; Pastor-Satorras & Castellano, 2016; Gromov, 2023) for these vectors: [4]

$$\text{IPR}_k^{(u)} := \left(\frac{\|\widetilde{U}_{k\cdot}\|_4}{\|\widetilde{U}_{k\cdot}\|_2}\right)^4; \qquad \text{IPR}_k^{(v)} := \left(\frac{\|\widetilde{V}_{k\cdot}\|_4}{\|\widetilde{V}_{k\cdot}\|_2}\right)^4; \qquad \text{IPR}_k^{(w)} := \left(\frac{\|\widetilde{W}_{\cdot k}\|_4}{\|\widetilde{W}_{\cdot k}\|_2}\right)^4; \qquad (3)$$

where $\|\cdot\|_P$ denotes the $L^P$-norm of the vector. One can readily see from equation 3 that $IPR \in [1/p, 1]$, with higher values for periodic vectors and lower values for non-periodic ones. It is useful to quantify the periodicity of each neuron in the hidden layer of the network (indexed by $k$). We define per-neuron IPR by averaging over the weight-vectors connected to the neuron:

$$\text{IPR}_k := \frac{1}{3}\left(\text{IPR}_k^{(u)} + \text{IPR}_k^{(v)} + \text{IPR}_k^{(w)}\right). \qquad (4)$$

Since trained networks generalize via the periodic features, a larger population of high-IPR neurons leads to better generalization. To quantify the overall similarity of the trained network to the analytical solution (equation 2) we average $\text{IPR}_k$ over all hidden neurons: $\overline{\text{IPR}} := \mathbb{E}_k\left[\text{IPR}_k\right]$.

---

[3]Periodicity of weights is approximate in trained networks.
[4]In general IPR is defined as $(\|U_{k\cdot}\|_{2r}/\|U_{k\cdot}\|_2)^{2r}$. We set $r = 2$, a common choice.

**Label corruption.** To introduce label corruption, we choose a fraction $\xi$ of the training examples, and replace the true labels with random labels. The corrupted labels are generated from a uniform random distribution over all the labels. This is often called *symmetric label noise* in literature (Song et al., 2022). This type of label corruption does not introduce label asymmetry.

## 1.2 RELATED WORKS

**Grokking** Grokking was first reported by Power et al. (2022) for Transformers trained on modular arithmetic datasets. Liu et al. (2022; 2023) presented explanations for grokking in terms of quality of representation learning as well as weight-norms. They also found grokking-like behaviours on other datasets. Nanda et al. (2023); Zhong et al. (2023) reverse-engineered grokked Transformer models and revealed the underlying learned algorithm. Barak et al. (2022) observed grokking on the sparse-parity problem, which they attributed to the amplification of the so-called Fourier gap. Merrill et al. (2023) examined grokking in the sparse-parity setup and attributed it to the emergence of sparse subnetworks. Žunkovič & Ilievski (2022) discuss solvable models for grokking in a teacher-student setup. Recently, Notsawo et al. (2023) investigated the loss landscape at early training time and used it to predict the occurrence of grokking. Varma et al. (2023) demonstrated new grokking-behaviours by investigating the critical amount of data needed for grokking.

**Label Noise** Zhang et al. (2017) showed that neural networks can easily fit random labels. A large body of work has been dedicated to developing techniques for robust training with label noise: explicit regularization(Hinton et al., 2012; Ioffe & Szegedy, 2015), noise-robust loss functions (Ghosh et al., 2017), sample selection (Katharopoulos & Fleuret, 2018; Paul et al., 2021), noise modeling and dedicated architectures (Xiao et al., 2015; Han et al., 2018; Yao et al., 2019) etc. We refer the reader to recent reviews by Song et al. (2022); Liang et al. (2022) for a comprehensive overview.

**Regularization and Pruning** Morcos et al. (2018) investigated the interplay between label corruption and regularization in vision tasks. They showed that BatchNorm (Ioffe & Szegedy, 2015) and Dropout (Hinton et al., 2012) have qualitatively different effects. Dropout effectively reduces the network-size but does not discourage over-reliance on a few special directions in weights space; while BatchNorm spreads the generalizing features over many dimensions. Bartoldson et al. (2020) found that there is a trade-off between stability and generalization when pruning trained networks.

## 2 GROKKING WITH LABEL NOISE

Grokking MLP on modular addition dataset is remarkably robust to label corruption. Even without explicit regularization, the model can generalize to near $100\%$ accuracy with sizable label corruption. In many cases, the network surprisingly manages to "correct" some of the corrupted examples, resulting in *Inversion* (*i.e.* test accuracy $>$ training accuracy). We emphasise that this is in stark contrast to the common belief that grokking requires explicit regularization. Adding regularization makes grokking more robust to label corruption, with stronger *Inversion*. To quantify these behaviours, we observe the training and test performance of the network with varying sizes of training dataset, amount of label corruption and regularization. We summarize our findings in the form of empirical phase diagrams (Figure 3), featuring the following phases. More detailed phase diagrams along with training and test performances are presented in Appendix L.

**Coexistence.** In the absence of explicit regularization the network generalizes on test data *and* memorizes the corrupted training data. In other words, both train and test accuracy are close to $100\%$ despite label corruption.

**Partial Inversion.** In this phase, the network generalizes on the test data but only memorizes a fraction of the corrupted training data. Remarkably, the network predicts the "true" labels on the remaining corrupted examples. In other words, the network *corrects* a fraction of the corrupted data. Consequently, we see $< 100\%$ train accuracy but near-$100\%$ test accuracy, resulting in a *negative* generalization gap (Figure 1(b))! We term this phenomenon *Partial Inversion*; where "inversion" refers to the test accuracy being higher than train accuracy. Remarkably, partial inversion occurs even in the absence of any explicit regularization, but only when there is ample training data (left-most panels in Figure 3(a,b)).

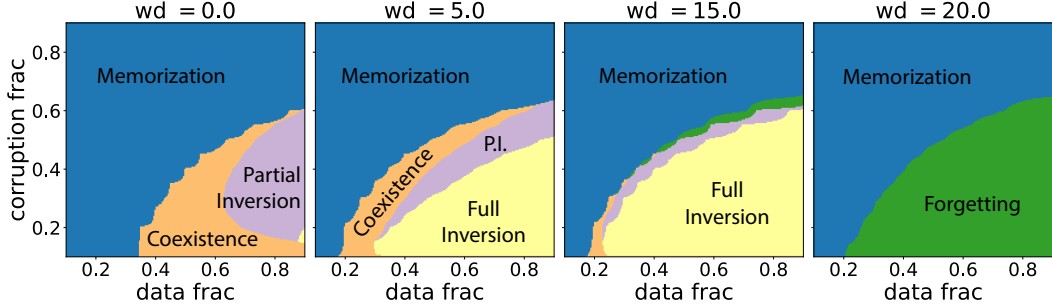

(a) Regularization with weight decay. (wd denotes the weight decay value)

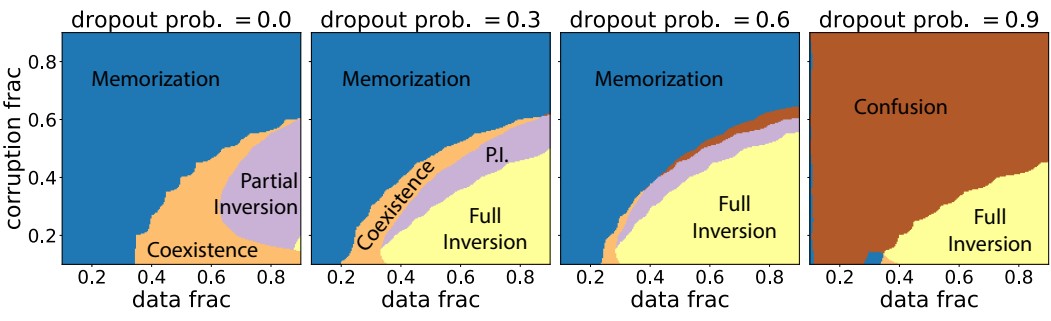

(b) Regularization with Dropout.

Figure 3: Modular Addition phase diagrams with various regularization methods. A larger data fraction leads to more "correct" examples, leading to higher corruption-robustness. Increasing regularization, in the form of weight decay or dropout, enhances robustness to label corruption and facilitates better generalization.

**Full Inversion.** Upon adding regularization, the network often generalizes on the test set but does not memorize any of the corrupted examples. This results in training accuracy plateauing close to $100(1 - \xi)\%$ ($\xi$ is the *corruption fraction*); while the test accuracy is at $100\%$ (Figure 1(c)). Increasing regularization enhances this phase (middle columns in Figure 3(a,b)).

**Memorization.** With very high label corruption and/or very low training data-fraction the network memorizes all of the training data (corrupted as well as non-corrupted), but does not generalize. We get close to $100\%$ accuracy on training data, but random guessing accuracy on test data in this phase.

**Forgetting.** With very high weight decay, the network performs poorly on both training and test data (right-most panel in Figure 3(a)). However, upon examination of the training curves (Figure 8(a)) we find that the accuracies plummet *after* the network has generalized! This performance drop is a result of the norms of weights collapsing to arbitrarily small values (Figure 8(b)). Forgetting exclusively occurs in grokked networks with activation functions of degree higher than 1 (e.g. we use $\phi(x) = x^2$) and with adaptive optimizers. (See Appendix E for a detailed discussion.)

**Confusion.** With very high dropout the network becomes effectively narrow, resulting in reduced capacity to memorize. Consequently, even in the absence of grokking, the network does not manage to memorize all the corrupted labels (right-most panel in Figure 3(b)).

## 2.1 GENERALIZING AND MEMORIZING SUB-NETWORKS

The generalizing features in our setup are "mechanistically" interpretable (equation 2). As a result, we can leverage measures of periodicity such as IPR to quantitatively characterize various phases presented above. It also allows us to analyze the effect of various regularization schemes. To that end, we state the following hypothesis and extensively examine it with experiments.

|  | Coexistence | Partial Inversion | Full Inversion | Memorization | Forgetting / Confusion |
|---|---|---|---|---|---|
| Memorizes corrupted train data | Yes | Partially | No | Yes | No |
| Generalizes on test data | Yes | Yes | Yes | No | No |

Table 1: Qualitative characterization of phases.

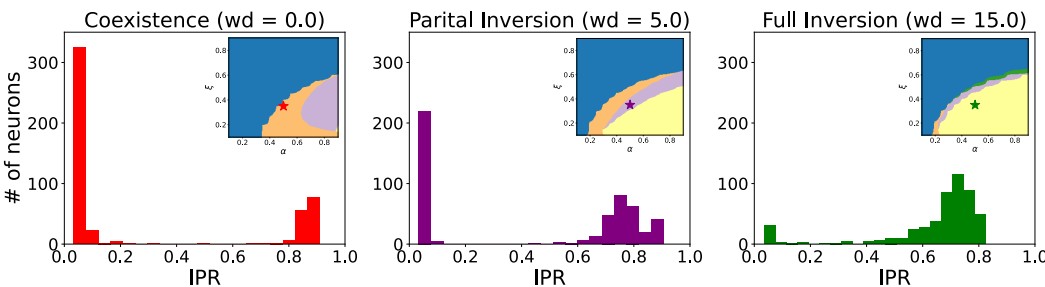

(a) Regularization with weight decay. (wd denotes the weight decay value)

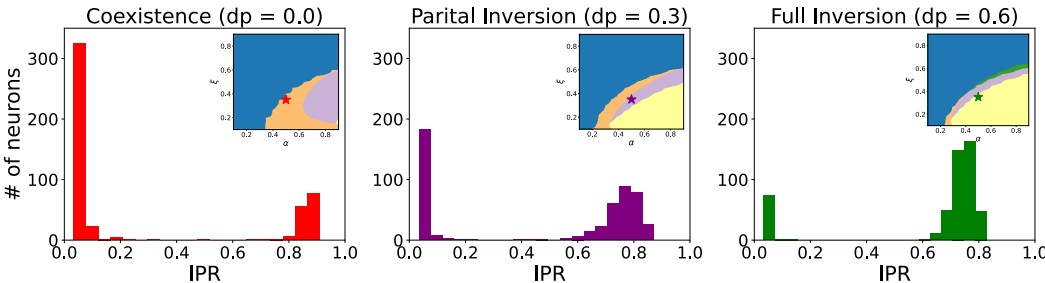

(b) Regularization with Dropout. (dp denotes the Dropout probability)

Figure 4: Distribution of per-neuron IPR for trained networks in various phases. Coexistence phase has a bimodal IPR distribution, where the high and low IPR neurons facilitate generalization and memorization, respectively. Regularization with weight decay or Dropout shifts the IPR distribution towards higher values – Generalizing neurons get more populous compared to memorizing ones; resulting in more robust generalization and *Inversion*. All plots are made for networks trained with data-fraction $\alpha = 0.5$ and corruption-fraction $\xi = 0.35$; with various regularization strengths.

**Hypothesis 2.1.** *For a two-layer MLP with quadratic activation trained on (one-hot encoded) modular addition datasets with label corruption, the high IPR neurons facilitate generalization via feature-learning whereas the low IPR neurons cause memorization of corrupted training data.*

To test Hypothesis 2.1 we study the distribution of per-neuron IPR in various phases, with different levels of regularization. In the *Coexistence* phase, the network simultaneously generalizes and memorizes the corrupted train data. From Hypothesis 2.1, we expect the presence of both generalizing and memorizing neurons. Indeed, we find a bi-modal distribution of IPRs in trained networks (left column of Figure 4). Remarkably, the network not only distinguishes between the tasks of feature-learning and memorization, it also assigns distinct and independent sub-networks to each.

As mentioned in the previous section, adding regularization in the form of weight decay or Dropout makes the generalization more robust and facilitates *Inversion*. Since these phases see decreased memorization of corrupted labels, we expect to see a decrease in the number of memorizing neurons and a corresponding increase in the generalizing ones. Indeed, we observe a distribution-shift towards higher IPR in the *Partial Inversion* phase (middle column of Figure 4). In the *Full Inversion* phase (induced by weight decay or dropout), we see that memorizing neurons are almost entirely

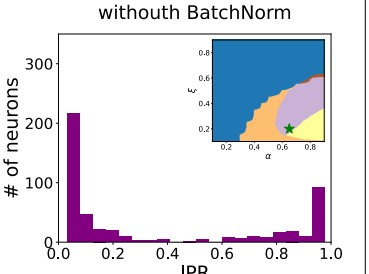 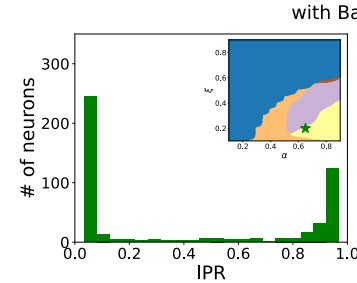 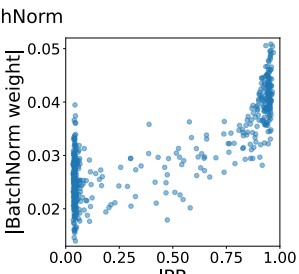

Figure 5: IPR distributions of networks with/without BatchNorm; the correlation between Batch-Norm weights and IPR. The IPR distributions in the two models do not exhibit significant differences. However, BatchNorm helps generalization by assigning higher weights ($\gamma_k$) to high IPR neurons compared to low IPR neurons. Both models are trained at data fraction $\alpha = 0.65$ and corruption fraction $\xi = 0.2$, with batch-size = 64.

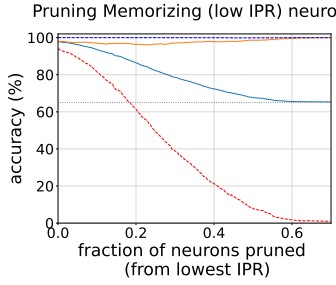 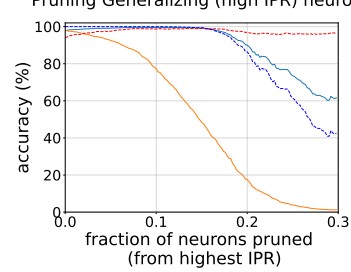 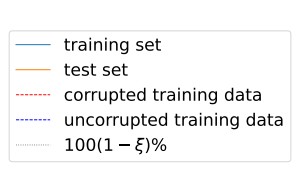

Figure 6: Progressively pruning neurons from a trained model (*Coexistence* phase), based on IPR. (data fraction $\alpha = 0.5$, corruption fraction $\xi = 0.35$). (Left) Pruning out neurons starting from the lowest IPR. The accuracy on corrupted training data decreases, providing evidence that low IPR neurons are responsible for memorization. Test accuracy as well as accuracy on uncorrupted training data remains high. (Right) Pruning out neurons starting from the highest IPR. Test accuracy decreases providing evidence that high IPR neurons are responsible for generalization. Accuracy on corrupted training data remains high.

eliminated in favour of generalizing ones (right column of Figure 4). This provides a mechanism by which the network recovers the "true" labels even for the corrupted data – in the absence of memorizing sub-network, its predictions are guided by the underlying rule and not corrupted samples.

**BatchNorm** also has a regularizing effect on training (Figure 9), albeit, through a qualitatively distinct mechanism. In the network (Equation (1)), we apply BatchNorm *after* the activation function (post-BN):

$$\boldsymbol{f}(m, n) = \boldsymbol{W} \, \mathrm{BN}\left(\phi\left(\boldsymbol{U}\,\boldsymbol{e}_m + \boldsymbol{V}\,\boldsymbol{e}_n\right)\right) . \tag{5}$$

BatchNorm layer does does not affect the IPR distribution of the neurons significantly – the low IPR neurons persist even in the *Full Inversion* phase (Figure 5). Instead, BatchNorm adjusts its own weights to de-amplify the low-IPR (memorizing) neurons, biasing the network towards generalizing representations. This can be seen from the strong correlation between $\mathrm{IPR}_k$ and the corresponding BatchNorm weights in Figure 5(right). BatchNorm effectively prunes the network and it learns to do this directly from the data! Note that this effect of BatchNorm acts in conjunction with the implicit regularization from the mini-batch noise in this case. We find that decreasing the batch-size enhances *Inversion* even without explicit regularization (Figure 10).

To further isolate the effect of individual neurons on performance, we perform the two complementary pruning experiments (Figure 6) : Starting from a trained network in the Coexistence phase, we gradually prune out neurons, one-at-a-time, (i) starting from the lowest IPR neuron (ii) starting from the highest IPR neuron, while keeping track of training and test accuracies. To distinguish network's ability to memorize, we also track the accuracies on corrupted and uncorrupted parts of the training dataset. Before pruning, the network has near-$100\%$ accuracy on both train and test data. In case

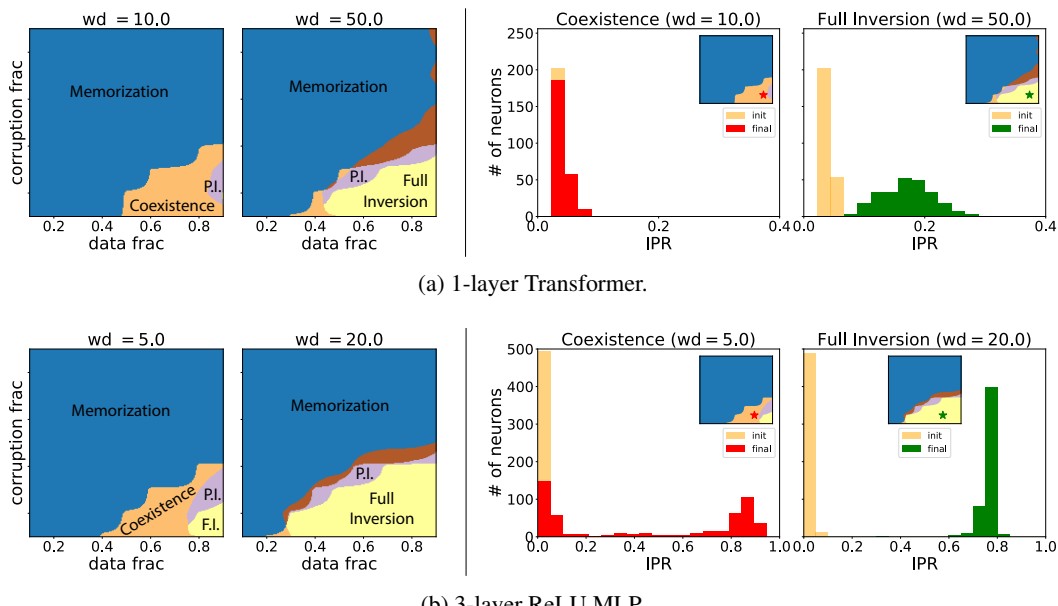

Figure 7: Modular Addition phase diagrams and IPR histograms with different architectures and weight decay values (denoted by "wd"). The IPR histograms are plotted for neurons connected to the input (embedding) layers.

(i) (Figure 6(left)), we see a monotonic decrease in accuracy on corrupted examples as more and more low IPR neurons are pruned. The accuracy on test data as well as ucorrupted training data remains high. The resulting (overall) training accuracy plateaus around $100(1 - \xi)\%$, corresponding to the fraction of uncorrupted examples. In case (ii) (Figure 6(right)), as more high IPR neurons are pruned, we see a decline in test accuracy (as well as the accuracy on the uncorrupted training data). The network retains its ability to memorize, so the accuracy on corrupted training data remains high.

We emphasize that, although each generalizing neuron learns the useful periodic features, the overall computation in the network is emergent: it is performed collectively by *all* generalizing neurons.

## 2.2 GENERAL ARCHITECTURES

In this subsection, we aim to check if (i) we can quantify memorization/generalization capabilities of general architectures using IPR distribution of the neurons; (ii) our understanding of regularization can be generalized to such cases.

**Hypothesis 2.2.** *For any network that achieves non-trivial generalization accuracy on the modular addition dataset with label corruption, its operation can be decomposed into two steps: (I) The input (embedding) layer with row-wise periodic weight maps the data onto periodic features; (II) The remainder of the network nonlinearly implements trigonometric operations over the periodic neurons and maps them to the output predictions. We hypothesize that step (I) depends only on the dataset characteristics; and hence, is a universal property across various network architectures.*

To verify Hypothesis 2.2 and better understand the impact of regularization, we conduct further experiments on 1-layer Transformers and 3-layer ReLU MLPs. Our findings, illustrated in the first two columns of Figure 7 and paralleling the observations in Figure 3(a), indicate that increased weight decay promotes generalization over memorization. IPR histograms in the last two columns of Figure 7 further confirm that weight decay encourages the correct features and corroborates the assertions made in Hypothesis 2.2.

For these general models, Dropout mostly leads to an extended confusion phase, while BatchNorm only facilitates coexistence. We observe weight decay to be essential for these models to have the *full inversion* phase. We refer the reader to Appendix G for an in-depth discussion.

## 3 EFFECT OF REGULARIZATION

In this section, we summarize the effect of various regularization techniques and compare them with existing literature.

### 3.1 WEIGHT DECAY

As shown in Figure 3(a), increasing weight decay has two notable effects: (i) Generalization with less data: The phase boundary between the *Memorization* and *Coexistence* phases shifts toward lower data-fraction. Weight decay encourages generalization even when training data is scarce; in line with classical ideas (Bartlett, 1998). (ii) Emergence of the *Full Inversion* phase: In this phase, weight decay prevents memorization and helps the network "correct" the labels for corrupted data points. This effect is further elucidated in Figure 4(a) – For a fixed amount of data and label corruption, increased weight decay results in diminished proportion of low IPR (memorizing) neurons in favor of high IPR (generalizing) ones.

### 3.2 DROPOUT

It is believed that dropout prevents complex co-adaptations between neurons in different layers, encouraging each neuron to learn useful features (Hinton et al., 2012; Srivastava et al., 2014). This intuition becomes more precise in our setup. The generalizing solution constitutes of periodic neurons, each corresponding to a weight-vector of a given "frequency" $\sigma(k)$. Such a solution requires $p$ neurons with different frequencies[5], (see Equation 6). In a wide network ($N = 500$ in our setup), there can be multiple neurons with the same frequency.[6] Due to this redundancy, generalizing solutions are more robust to dropped neurons; whereas overfitting solutions are not. As a result, training with Dropout gets attracted to such periodic solutions.

## 4 CONCLUSION

We have investigated the interplay between memorization and generalization in two-layer neural networks trained on algorithmic datasets. The memorization was induced by creating a dataset with corrupted labels, forcing the network to memorize incorrect examples and learn the rule at the same time. We have found that the memorizing neurons can be explicitly identified and pruned away leading to perfect generalization.

Next we showed that various regularization methods such as weight decay, Dropout and BatchNorm encourage the network to ignore the corrupted labels and lead to perfect generalization. We have leveraged the interpretability of the representations learnt on the modular addition task to quantitatively characterize the effects of these regularization methods. Namely, we have found that the weight decay and dropout act in a similar way by reducing the number of memorizing neurons; while BatchNorm de-amplifies the output of regularizing neurons without eliminating them, effectively pruning the network.

## 5 LIMITATIONS AND FUTURE WORK

Although the trained models in our setup are completely interpretable by means of analytical solutions, the training dynamics that lead to these solutions remains an open question. Solving the dynamics should also shed more light on the effects of regularization and label corruption.

While our work is limited to the exactly solvable models and algorithmic datasets, it paves the way to quantitative investigation of generalization and memorization in more realistic settings.

---

[5]More precisely, it only requires $\lceil p/2 \rceil$ frequencies for real weights.

[6]The analytical solution becomes exact only at large $N$; with sub-leading corrections of order $1/\sqrt{N}$.

## ACKNOWLEDGMENTS

A.G.'s work at the University of Maryland was supported in part by NSF CAREER Award DMR-2045181, Sloan Foundation and the Laboratory for Physical Sciences through the Condensed Matter Theory Center. The authors acknowledge the University of Maryland supercomputing resources (http://hpcc.umd.edu) made available for conducting the research reported in this paper.

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

## A   FURTHER EXPERIMENTAL DETAILS

In every plot presented, unless explicitly stated otherwise, we use the following default setting:

**Default setting**   A 2-layer fully connected network with quadratic activation function and width $N = 500$, initial weights sampled from $\mathcal{N}(0, (16N)^{-2/3})$, trained on $p = 97$ Modular Addition dataset. The networks are trained for 2000 optimization steps, using full-batch, MSE loss, AdamW optimizer, learning rate $\eta = 0.01$ and $\beta = (0.9, 0.98)$.

### A.1   FIGURES IN MAIN TEXT

Figure 1: Default setting with $\alpha = 0.5, \xi = 0.35$ and (a) $wd = 0$ (b) $wd = 5$ and (c) $wd = 15$.

Figure 2: Default setting with $\alpha = 0.5, \xi = 0.0, wd = 5.0$.

Figure 3: Each plot is scanned over 17 different data fractions $\alpha$ ranging from 0.1 to 0.9 and 19 noise levels $\xi$ from 0.0 to 0.9 using the default setting.

Figures 4 and 6: Default setting with $\alpha = 0.5, \xi = 0.35$.

Figure 5: 2-layer fully connected network with quadratic activation function and width $N = 500$, with and without BatchNorm. We train for $\sim 8000$ steps, with MSE loss, Adam optimizer with minibatch of size 64, and learning rate $\eta = 0.005$. Data fraction $\alpha = 0.65$, corruption fraction $\xi = 0.2$.

General setting for minibatch training: We train the network with batch size 256 for 2000 steps. For batch sizes 8 and 64, we scale the number of total steps so as to keep the number of epochs roughly equal. (In other words, all the networks see the same amount of data.) The learning rate is scaled as $0.01 \cdot \sqrt{\text{batch size}/256}$.

Figure 7: All networks are trained with MSE loss using AdamW optimizer with learning rate $\eta = 0.001$ and $(\beta_1, \beta_2) = (0.9, 0.98)$. (a) Encoder-only Transformer, embedding dimension $d_{\text{embed}} = 128$ with ReLU activation, trained for 3000 steps with 3 random seeds; (b) Width $N = 500$ trained for 5000 steps with 3 random seeds.

## B   DETAILS OF RESOURCES USED

Figure 3 requires 72 hours on a single NVIDIA RTX 3090 GPU.

Figure 7 used 60 hours on a 1/7 NVIDIA A100 GPU.

All the remaining figures except Figure 7 in the main text are either further experiments performed on the data from the Figure 3 experiment; or are very quick to perform on any machine.

We used  300 GPU hours for early-stage exploration and making phase diagrams in the appendix.

## C  QUANTITATIVE CHARACTERIZATION OF PHASES

All phase diagrams are plotted using the empirical accuracies of trained networks.

Trained networks with $\geq 90\%$ test accuracies get classified into *Coexistence*, *Partial Inversion* and *Full Inversion* based on their training accuracy (specifically, their ability to memorize corrupted examples). *Coexistence* phase has $\geq 90\%$ training accuracy, since it memorizes most of the corrupted training examples. *Partial Inversion* phase has $< 90\%$ but $\geq 100(1.05 - \xi)\%$ training accuracy, where $\xi$ is the corruption fraction. Note that we subtract $\xi$ from $1.05$ instead of $1.00$ to account for the $1 - 2\%$ random guessing accuracy as well as statistical fluctuations around this value. *Full Inversion* phase has $< 100(1.05 - \xi)\%$ training accuracy, since it does not memorize any of the corrupted examples. *Memorization* phase has $< 90\%$ test accuracy but $\geq 90\%$ training accuracy, indicating that the network memorizes all of the training data, including the corrupted and uncorrupted examples. *Forgetting* or *confusion* phase has $< 90\%$ test as well as training accuracies. the network neither generalizes nor manages to memorize all the training examples. We distinguish *Forgetting* from *confusion* using the training curve. In the *Forgetting phase* the grokked network loses its performance due to steadily decreasing weight-norms. Whereas in the *confusion* phase, the network never reaches high performance.

|  | **Coexistence** | **Partial Inversion** | **Full Inversion** | **Memorization** | **Forgetting / Confusion** |
|---|---|---|---|---|---|
| Test accuracy | $\geq 90\%$ | $\geq 90\%$ | $\geq 90\%$ | $< 90\%$ | $< 90\%$ |
| Training accuracy | $\geq 90\%$ | $< 90\%$ and $\geq 100(1.05 - \xi)\%$ | $< 100(1.05 - \xi)\%$ | $\geq 90\%$ | $< 90\%$ |

Table 2: Quantitative characterization of phases.

# D  ANALYTICAL SOLUTION

We show that the network with periodic weights stated in Equation (2) is indeed the analytical solution for the modular arithmetic task. The network output is given by

$$
\begin{aligned}
f_q(m, n) &= \sum_{k=1}^{N} W_{qk}(U_{km} + V_{kn})^2 \\
&= \frac{2}{N} \sum_{k=1}^{N} \cos\left(-\frac{2\pi}{p}\sigma(k)q - (\phi_k^{(u)} + \phi_k^{(v)})\right) \cdot \\
&\qquad\qquad \cdot \left(\cos\left(\frac{2\pi}{p}\sigma(k)m + \phi_k^{(u)}\right) + \cos\left(\frac{2\pi}{p}\sigma(k)n + \phi_k^{(v)}\right)\right)^2 \\
&= \frac{2}{N} \sum_{k=1}^{N} \left\{ \frac{1}{4}\cos\left(\frac{2\pi}{p}\sigma(k)(2m - q) + \phi_k^{(u)} - \phi_k^{(v)}\right) \right. \\
&\qquad + \frac{1}{4}\cos\left(\frac{2\pi}{p}\sigma(k)(2m + q) + 3\phi_k^{(u)} + \phi_k^{(v)}\right) \\
&\qquad + \frac{1}{4}\cos\left(\frac{2\pi}{p}\sigma(k)(2n - q) - \phi_k^{(u)} + \phi_k^{(v)}\right) \\
&\qquad + \frac{1}{4}\cos\left(\frac{2\pi}{p}\sigma(k)(2n + q) + \phi_k^{(u)} + 3\phi_k^{(v)}\right) \\
&\qquad + \boxed{\frac{1}{2}\cos\left(\frac{2\pi}{p}\sigma(k)(m + n - q)\right)} \\
&\qquad + \frac{1}{2}\cos\left(\frac{2\pi}{p}\sigma(k)(m + n + q) + 2\phi_k^{(u)} + 2\phi_k^{(v)}\right) \\
&\qquad + \frac{1}{2}\cos\left(\frac{2\pi}{p}\sigma(k)(m - n - q) - 2\phi_k^{(v)}\right) \\
&\qquad + \frac{1}{2}\cos\left(\frac{2\pi}{p}\sigma(k)(m - n + q) + 2\phi_k^{(u)}\right) \\
&\qquad \left. + \cos\left(-\frac{2\pi}{p}\sigma(k)q - \phi_k^{(u)} - \phi_k^{(v)}\right) \right\}.
\end{aligned}
\tag{6}
$$

We have highlighted the term that will give us the desired output with a box. Note that the desired term is the only one that does not have additive phases in the argument. Recall that the phases $\phi_k^{(u)}$ and $\phi_k^{(v)}$ randomly chosen – uniformly i.i.d. sampled between $(-\pi, \pi]$. Consequently, as $N$ becomes large, all other terms will vanish due to random phase approximation. The only surviving term will be the boxed term. We can write this boxed term in a more suggestive form to make the analytical solution apparent.

$$
f_q(m, n) = \frac{1}{N} \sum_{k=1}^{N} \cos\left(\frac{2\pi}{p}\sigma(k)(m + n - q)\right) \approx \delta^p(m + n - q),
\tag{7}
$$

where we have defined the *modular Kronecker Delta function* $\delta^p(\cdot)$ as Kronecker Delta function up to integer multiples of the modular base $p$.

$$
\delta^p(x) = \begin{cases} 1 & x = rp \quad (r \in \mathbb{Z}) \\ 0 & \text{otherwise} \end{cases},
\tag{8}
$$

where $\mathbb{Z}$ denotes the set of all integers. Note that $\delta^p(m + n - q)$ are the desired `one_hot` encoded labels for the Modular Arithmetic task, by definition. Thus our network output with periodic weights is indeed a solution.

# E   FORGETTING PHASE

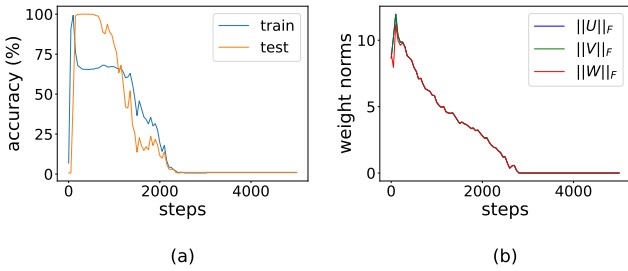

(a)                                                    (b)

Figure 8: *Forgetting* phase (data fraction $\alpha = 0.5$, corruption fraction $\xi = 0.35$, weight decay = 20). (a) Training curves. (b) Frobenius norms of network weights with training.

We saw in Figure 3(a) (right-most panel, wd=20.0) that extremely high weight decay leads to *Forgetting* phase. This phase has random guessing accuracies on both training and test datasets at the end of training. Naively, this would seem similar to the *Confusion* phase. However, Upon close inspection of their training dynamics, we find rich behaviour. Namely, the network initially undergoes dynamics similar to *Full Inversion*, *i.e.*, the test accuracy reaches $100\%$ while the train accuracy plateaus around $100(1-\xi)\%$. However, both accuracies subsequently decay down to random guessing. This curious phenomenon is explained by the effect of the high weight decay on the network weights. We notice that the Frobenius norms of the weights of the network steadily decay down to arbitrarily small values.

This curious behaviour only occurs when we use activation functions of degree higher than 1 (e.g. $\phi(x) = x^2$ in this case). (More precisely, we mean that the first term in the series expansion should be of degree at least 2.) This can be intuitively explained by examining the gradients carefully. With quadratic activation functions and MSE loss, gradients take the following form

$$\frac{\partial \mathcal{L}}{\partial U_{ab}} = \frac{4}{|D|p} \sum_{(b,n) \in D} \sum_{q}^{p} \left( \sum_{k=1}^{N} W_{qk} \left( U_{kb} + V_{kn} \right)^2 - \delta^p(b+n-q) \right) W_{qa} \left( U_{ab} + V_{an} \right)$$

$$\frac{\partial \mathcal{L}}{\partial V_{ab}} = \frac{4}{|D|p} \sum_{(m,b) \in D} \sum_{q}^{p} \left( \sum_{k=1}^{N} W_{qk} \left( U_{km} + V_{kb} \right)^2 - \delta^p(b+n-q) \right) W_{qa} \left( U_{am} + V_{ab} \right) \quad (9)$$

$$\frac{\partial \mathcal{L}}{\partial W_{cd}} = \frac{4}{|D|p} \sum_{(m,n) \in D} \left( \sum_{k=1}^{N} W_{ck} \left( U_{km} + V_{kn} \right)^2 - \delta^p(m+n-q) \right) \left( U_{dm} + V_{dn} \right)^2 \ ,$$

where $D$ is the training dataset and $|D|$ is its size.

For non-generalizing solutions, the term inside the bracket is of degree 0 in weights, due to the delta function term. But close to generalizing solutions, the term inside the bracket becomes effectively degree 3 in weights, since the delta function term cancels with the network output. Consequently, around the generalizing solution, the weight gradients have a higher degree dependence on the weights themselves. This makes such solutions more prone to the collapse due to high weight decay. This serves as a heuristic justification as to why only grokked networks seemed to be prone to *Forgetting*, while memorizing networks seem to be immune.

If we use vanilla Gradient Descent optimizer, the updates would plateau once the balance between gradient updates and weight decay contribution is reached. But this is not the case in Adaptive optimizers. Indeed, we find that this phenomenon exclusively occurs with Adaptive optimizers (e.g. Adam in this case).

We believe that it is possible to get further insight into this phase by closely examining the optimization landscape and network dynamics. Since this behaviour is exclusive to grokked networks, understanding this phase would lead to further understanding of grokking dynamics. Alternatively, a better understanding of the dynamics of grokking should give us insights into this curious phenomenon. We defer this analysis for future work and welcome an examination by our readers.

# F  COMPARISON OF BATCHNORM, LAYERNORM AND MINIBATCH W/O NORMALIZATION

Here we examine the effects of BatchNorm, LayerNorm and mini-batch noise. We find that generalization and *Inversion* are indeed enhanced by decreasing the batch-size. This demonstrates that implicit regularization from mini-batch noise favours simpler solutions in our setup.

Additionally, upon comparing Figure 9, Figure 10 and Figure 11 we find enhanced *Inversion* in cases with normalization layers (especially BatchNorm). For BatchNorm, we provide an explanation in the main text, with the the BatchNorm weights de-amplifying memorizing neurons in favour of generalizing ones. For LayerNorm, however, we do not find such an effect.

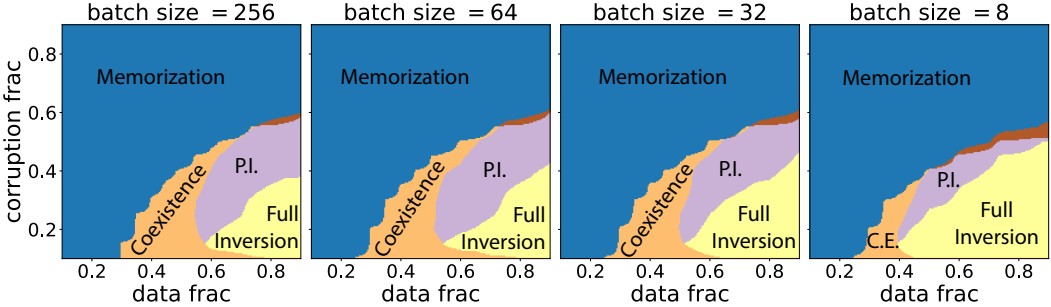

Figure 9: Phase diagram with BatchNorm (post-BN)

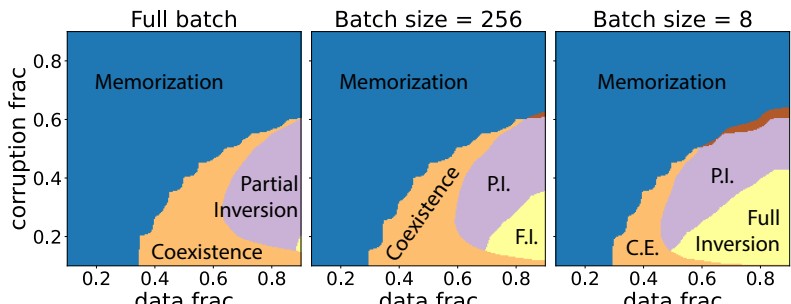

Figure 10: Phase diagram with mini-batch training, without any explicit regularization or normalization layers.

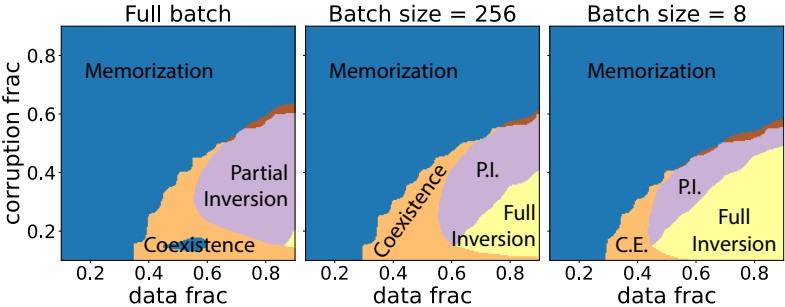

Figure 11: Phase diagram with LayerNorm (post-LN)

## G  TRANSFORMERS AND DEEP MLPS

In this section, we present (i) more weight decay experiments on different configurations of Transformers; (ii) the Effect of Dropout on Transformers; (iii) the Effect of BatchNorm on 3-layer ReLU MLPs.

For reasons that we do not fully understand, 3-layer ReLU MLPs with even $0.05$ Dropout probabilities failed to memorize the dataset for any data fraction $\alpha > 0.3$ and noise level $\xi \geq 0.1$, even after an extensive hyperparameter search. We leave the discussion about this for the future.

### G.1  EFFECT OF DEPTH / NUMBER OF HEADS / WEIGHT TYING

To further support our Hypothesis 2.2, we conducted further experiments by varying the configuration of Transformer models. Figure 12 suggests that our Hypothesis holds irrespective of the Transformer configurations.

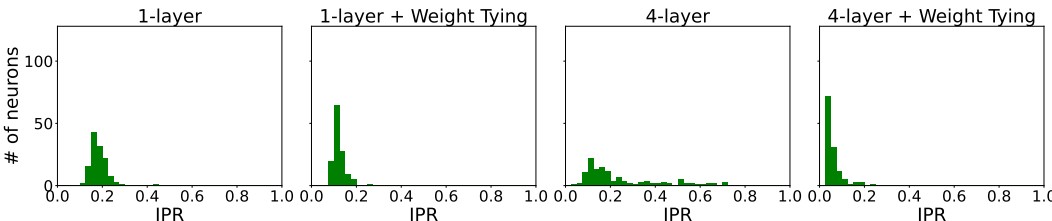

(a) Transformer with 8 heads and embedding dimension 128.

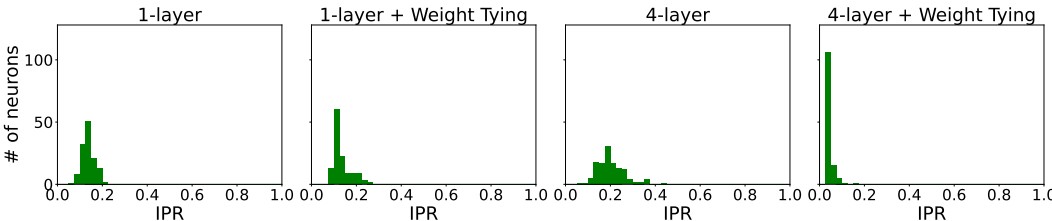

(b) Transformer with 4 heads and embedding dimension 128.

Figure 12: The IPR histograms of the Embedding layer for Transformers with different configurations. We see that depth does not affect the IPR distribution, whereas more heads encourages high IPR embeddings and weight tying discourages it. All models are trained with corruption fraction $\alpha = 0.8$ and corruption fraction $\xi = 0.2$.

### G.2  DROPOUT PHASE DIAGRAM FOR TRANSFORMERS

We conduct further experiments to check the effect of Dropout on Transformers. From Figure 13, we see that with a small dropout probability, the model managed to have a coexistence phase with an average higher IPR for neurons connected to the embedding layer. Notably, any dropout probability larger than $0.1$ does not help.

### G.3  EFFECT OF PARAMETERIZATION FOR 3-LAYER MLPS

As we mentioned in Hypothesis 2.2, the input layer should have high IPR columns irrespective of the details of the network. In this section, we show that different parameterization of the network does not break our Hypothesis.

In particular, we consider two different parameterizations: (i) The standard parameterization with weights sampled from $\mathcal{N}(0, 2/\text{fan\_in})$; (ii) The parameterization with weights sampled from $\mathcal{N}(0, 2/N^{2/3})$, which corresponds to a network parameterized in a more lazy training regime (Jacot et al., 2022).

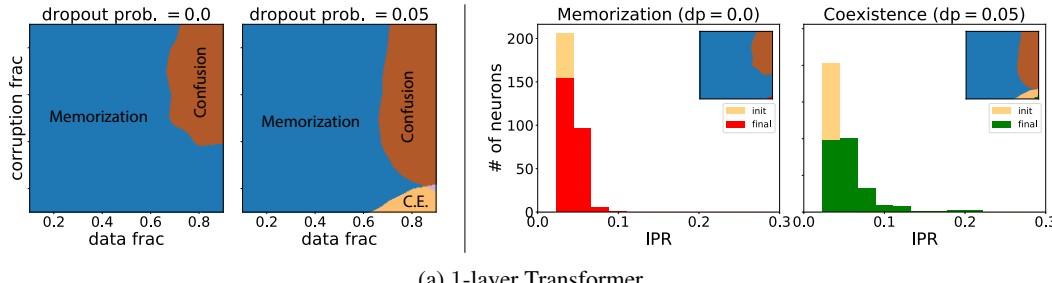

(a) 1-layer Transformer.

Figure 13: Modular Addition phase diagrams and IPR histograms with different dropout probabilities (denoted by "dp"). The IPR histograms are plotted for neurons connected to the embedding layer. We select points trained with data fraction $\alpha = 0.9$ and corruption fraction $\xi = 0.1$.

Our results in Figure 14 show that as the network gets into a lazier regime, the row-wise IPR distribution of the input layer weight barely shifts, whereas the column-wise IPR distribution of the output layer weight has larger values. We interpret this shift as happening because the network with lazier parameterization fails to utilize the hidden layers to decode the periodic features embedded by input layers.

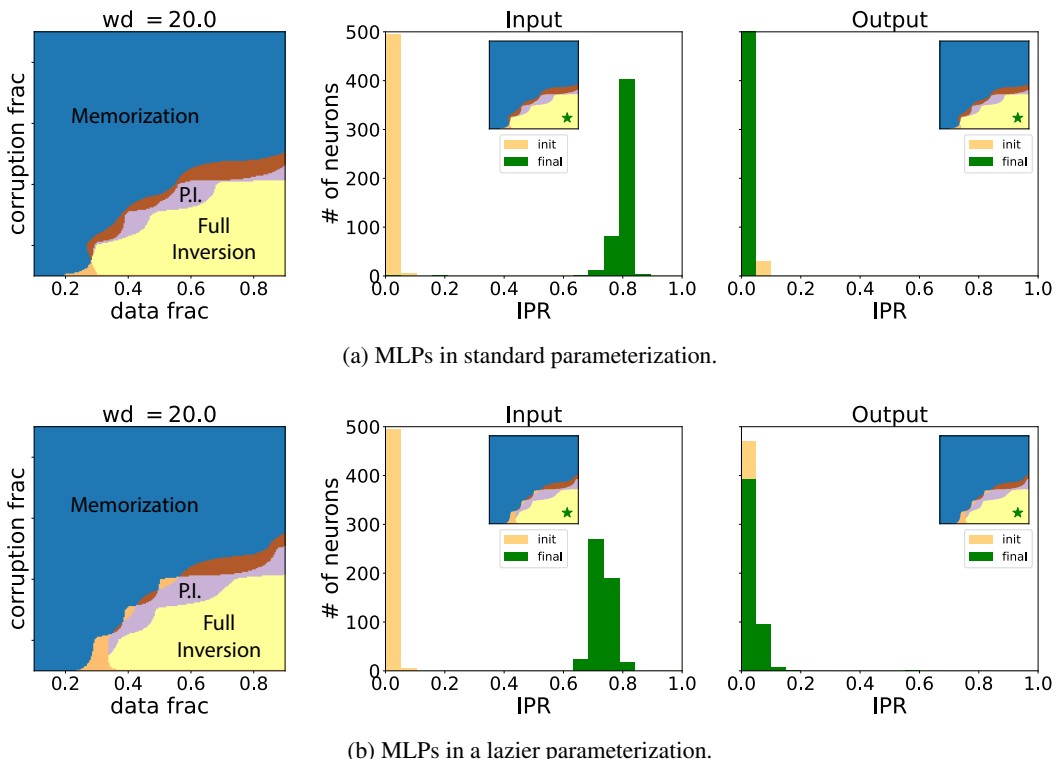

(a) MLPs in standard parameterization.

(b) MLPs in a lazier parameterization.

Figure 14: Phase diagrams and the IPR histograms of the columns of input layer weight and rows of the output layer weights for 3-layer MLPs with different parameterizations. Selected models are trained with data fraction $\alpha = 0.8$ and corruption fraction $\xi = 0.2$.

## G.4 BATCHNORM FOR 3-LAYER RELU MLPS

This subsection tests the correlation between BatchNorm weights and high IPR neurons for 3-layer ReLU MLPs. We found that BatchNorm for 3-layer MLPs only has coexistence phases and can perfectly memorize and generalize for small corruption fractions $\xi$, we showed a representative training curve in Figure 15. We find that with BatchNorm, the 3-layer MLP tends to have high IPR

neurons close to the output layer instead of having high IPR for neurons close to the input layer. We believe that failure to remove the very low IPR neurons connected to the input layer is why this network is always in the coexistence phase.

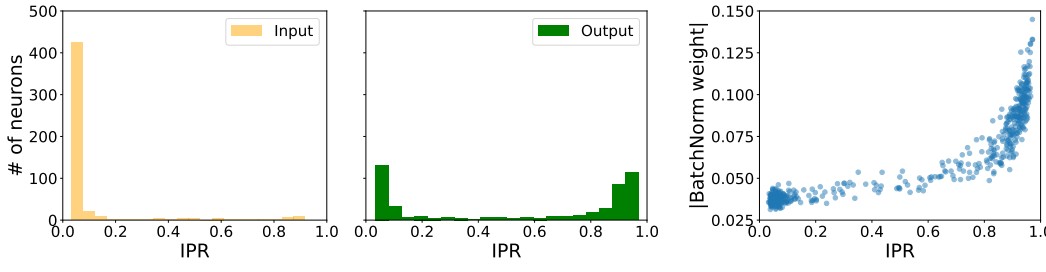

(a) 3-layer ReLU MLP, trained with batch size $B = 64$.

Figure 15: IPR of a 3-layer MLP in coexistence phase, trained with data fraction $\alpha = 0.8$ and corruption fraction $\xi = 0.1$. We plotted row-wise IPR for input layer weight and column-wise IPR for output layer weight; the correlation between column-wise IPR of output layer weights with the absolute value of BatchNorm weights right before it. The correlation between the BatchNorm weights and IPR values we pointed out in the main text still holds. However, we did not observe a strong correlation between row-wise IPR of the input layer and the BatchNorm weight next to it.

## H  LABEL CORRUPTION ON GENERAL DATASETS

We train a 4-layer ReLU MLP on the MNIST dataset with label corruption with weight decay and dropout (CrossEntropy loss). Weight decay indeed enhances generalization and leads to *Full Inversion*. Very high weight decay leads to confusion. Dropout, on the other hand, is not as effective. Increasing dropout hurts both test *and* train accuracies in this case.

The generalizing features are harder to quantify for MNIST – we leave this analysis for future work.

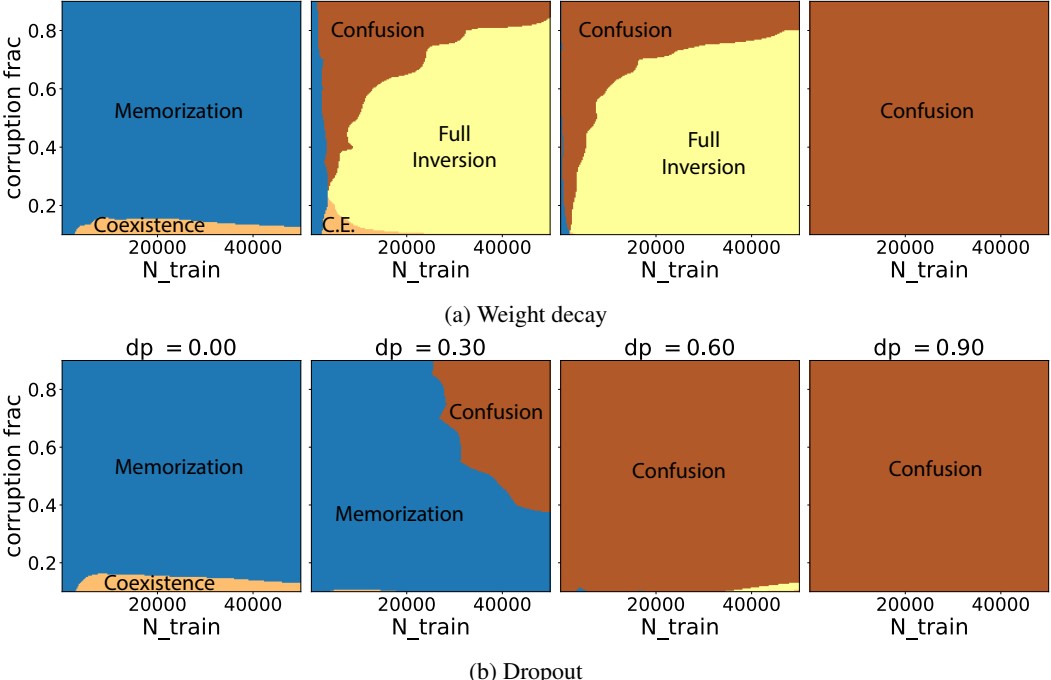

Figure 16: 4-layer ReLU MLP trained on MNIST dataset with label corruption.

# I   ADDITIONAL PRUNING CURVES

## I.1   ACCURACIES

In Figure 6, we showed pruning low and high IPR neurons in the *Coexistence* phase. Here we show the extended version of the pruning experiments. In contrast to the plots in Figure 6, In Figure 17 we plot longer x-axes – we show accuracies with fraction of pruned neurons ranging from 0.0 to 1.0. We also perform the pruning experiment for *Partial Inversion* and *Full Inversion* phases. Pruning affects generalization and memorization differently in different phases because the proportion of high and low IPR neurons in trained networks depends on the phase.

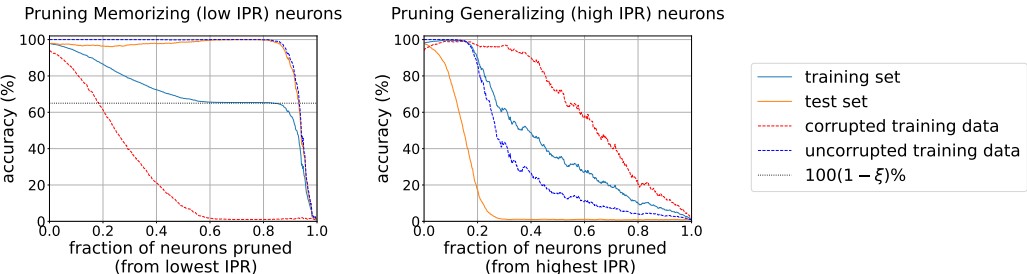

(a) *Coexistence* phase (weight decay = 0.0). (Left) Pruning from low IPR neurons retains performance on test data and uncorrupted training data; while decreases performance on corrupted training data down to to $100 * (1 - \xi)\%$. (Right) Pruning high IPR neurons decreases performance on test data; while retaining performance on training data. Note that the accuracy on corrupted data is retained longer than that on uncorrupted data.

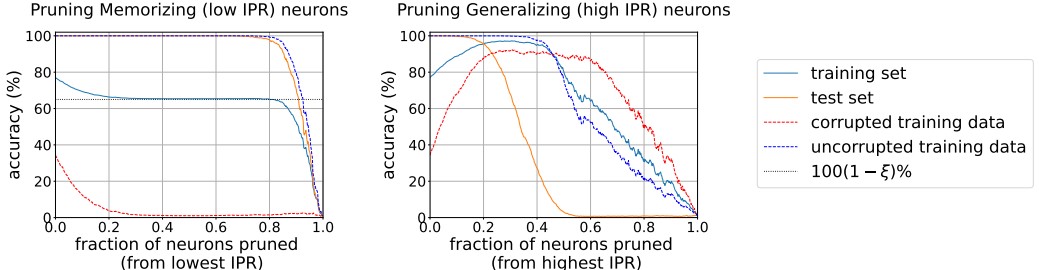

(b) *Partial Inversion* phase (weight decay = 5.0). (Left) Pruning from low IPR neurons retains performance on test data and uncorrupted training data; while decreases performance on corrupted training data down to to $100 * (1 - \xi)\%$. (Right) Pruning high IPR neurons decreases performance on test data. It improves the performance on corrupted training data and retains it on uncorrupted training data. In the absence of periodic neurons, the (memorizing) low-IPR neurons have an increased influence on the network predictions, which leads to this increase in memorization.

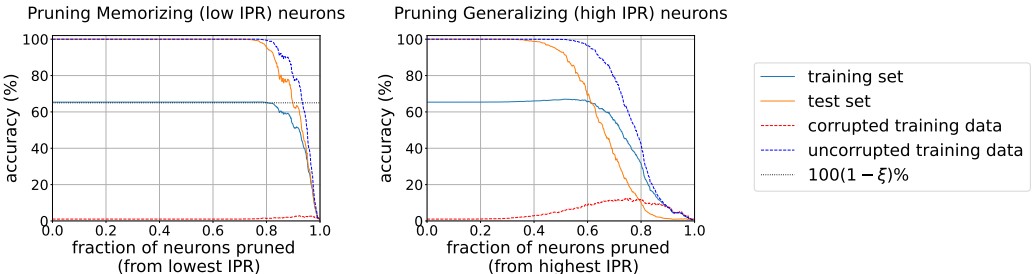

(c) *Full Inversion* phase (weight decay = 15.0). (Left) Pruning from low IPR neurons retains performance on test data and uncorrupted training data; while decreases performance on corrupted training data to $100 * (1 - \xi)\%$. (Right) Pruning high IPR neurons does not improve memorization significantly, except for the small increase in accuracy on corrupted training data towards the end. This is because the fraction of the neurons that have low IPR is very small in this case.

Figure 17: Pruning experiments in all various phases ($N = 500, \alpha = 0.5, \xi = 0.35$). Pruning affects generalization and memorization differently in different phases because the proportion of high and low IPR neurons in trained networks depends on the phase.

## I.2 LOSSES

Here we show how the train and test losses are affected by pruning. Train loss is lowered by, both, memorizing (low IPR) *and* generalizing (high IPR) neurons. This leads to monotonic behaviors in all the cases. Test loss is lowered by periodic neurons, while increased by memorizing neurons. An effect that sits on top of these is the effective cancellation of sub-leading terms in the analytical solution Equation (6). Since the accuracies only depend on the largest logit, these sub-leading terms have little effect on them (at sufficient width). However, losses are significantly affected by them. With zero or small weight decay values, this cancellation is achieved by low IPR neurons. Thus, in these cases, low IPR neurons serve the dual purpose of memorization as well as canceling the sub-leading terms. With large weight decay values, this cancellation is achieved by a different algorithm, where "secondary peaks" appear in the Fourier spectrum of the periodic neurons. We leave an in-depth analysis of these algorithmic biases for future work.

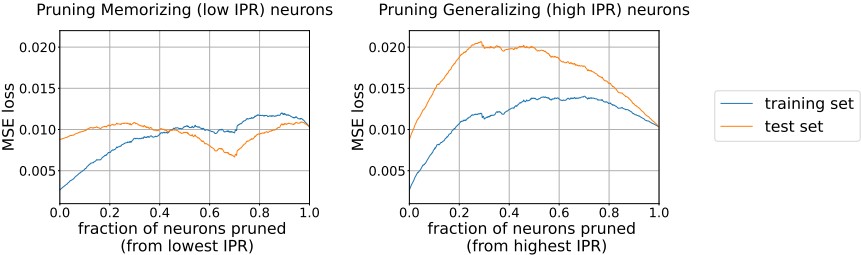

(a) *Coexistence* phase (weight decay = 0.0). (Left) Pruning from low IPR neurons (almost) monotonically increases the train loss. The test loss has two competing factors: lower memorization and fewer neurons that cancel the sub-leading terms. As a result, we see non-monotonic behaviour in the first half. (Right) Pruning high IPR neurons increases both train and test losses. We see a non-monotonic behaviour in the second half since the cancellation of sub-leading terms is irrelevant in the absence of periodic neurons.

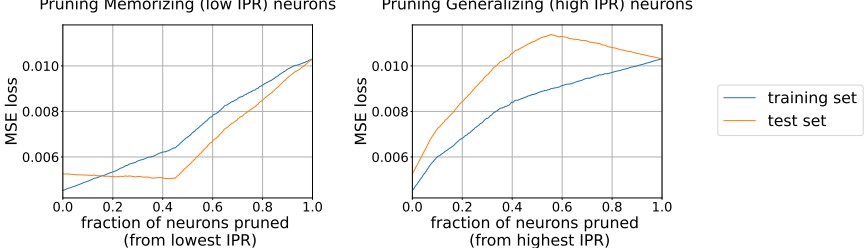

(b) *Partial Inversion* phase (weight decay = 5.0). (Left) Pruning from low IPR neurons: Initially, the removal of memorizing neurons causes the train loss to increase and the test loss to decrease. In the latter half, once we start removing the periodic neurons, both train and test losses increase. (Right) Pruning from high IPR neurons: Initially, both train and test losses increase due to the removal of periodic neurons. In the latter half, the removal of the memorizing neurons causes the train and test losses to increase and decrease, respectively.

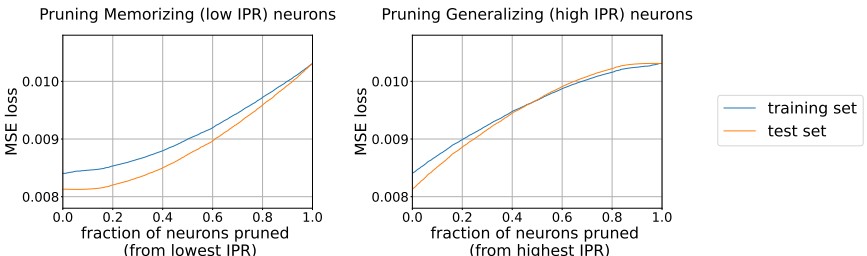

(c) *Full Inversion* phase (weight decay = 15.0). (Left) Pruning from low IPR neurons monotonically increases the train loss, since it is affected by removal of memorizing *and* periodic neurons. On the other hand, test loss briefly remains constant (during the removal of the low IPR neurons), followed by a monotonic increase. (Right) Removing high IPR neurons monotonically increases both train and test losses.

Figure 18: Pruning experiments in all various phases ($N = 500, \alpha = 0.5, \xi = 0.35$). Pruning affects generalization and memorization differently in different phases because the proportion of high and low IPR neurons in trained networks depends on the phase.

## J ADDITIONAL TRAINING CURVES

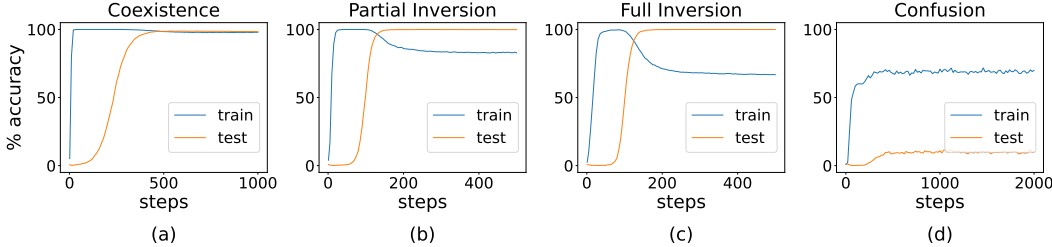

Figure 19: Training curves with dropout (a) dp=0.0 (b) dp=0.3 (c) dp=0.6 (d) dp=0.9

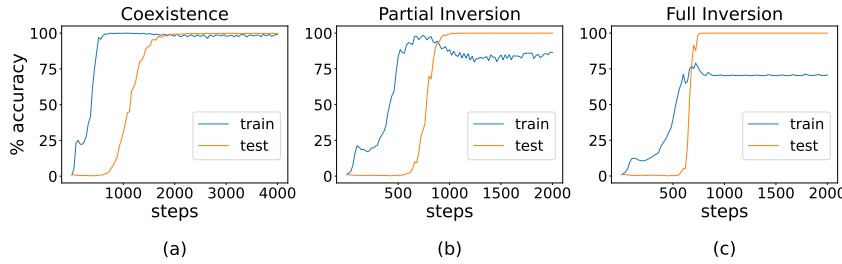

Figure 20: Training curves in various phases with BatchNorm. All networks are trained with corruption fraction $\xi = 0.3$ and batch-size = 64; with variable data fractions. (a) Coexistence ($\alpha = 0.4$) (b) Partial Inversion ($\alpha = 0.5$) (c) Full Inversion ($\alpha = 0.8$)

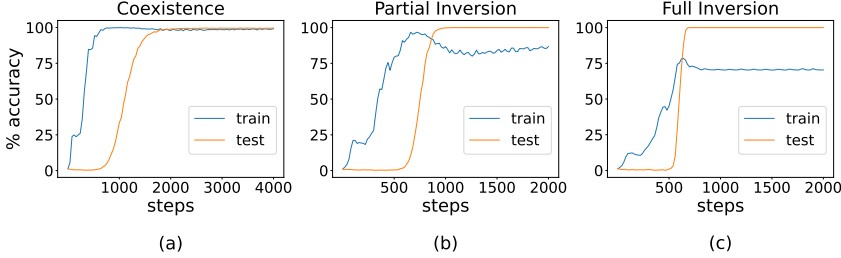

Figure 21: Training curves in various phases with LayerNorm. All networks are trained with corruption fraction $\xi = 0.3$ and batch-size = 64; with variable data fractions. (a) Coexistence ($\alpha = 0.4$) (b) Partial Inversion ($\alpha = 0.5$) (c) Full Inversion ($\alpha = 0.8$)

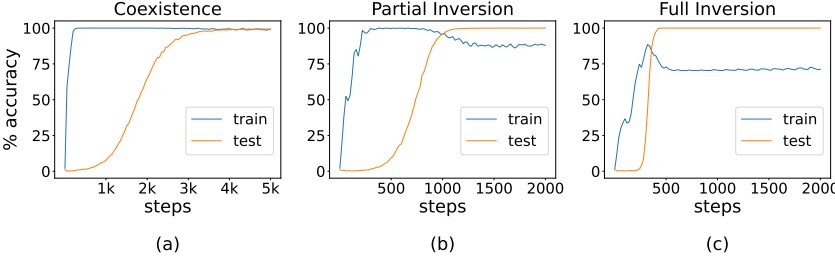

Figure 22: Training curves in various phases without any normalization, but with mini-batches. All networks are trained with corruption fraction $\xi = 0.3$ and batch-size = 64; with variable data fractions. (a) Coexistence ($\alpha = 0.4$) (b) Partial Inversion ($\alpha = 0.5$) (c) Full Inversion ($\alpha = 0.8$)

# K OUTPUT LOGITS

Here we show how the output logits of trained networks look like in various phases (i.e. different amount of weight decay) on corrupted training examples. In all cases, we see two peaks in logits, corresponding the the corrupted and the "correct" labels. Their relative value determines the degree of memorization, and hence, the phase.

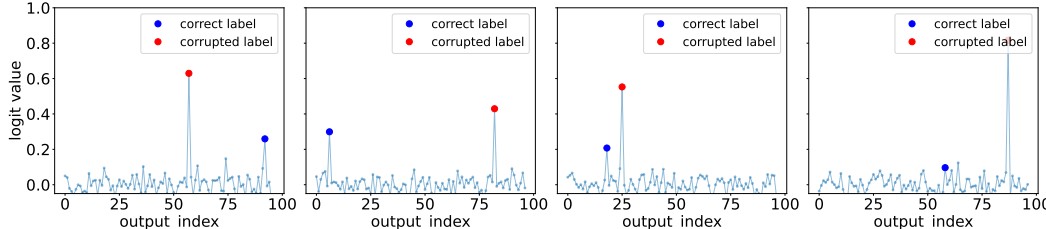

(a) *Coexistence* phase (wd=0.0). The logits corresponding to corrupted label have a higher values than the correct ones, which explains the memorization of corrupted data in this phase.

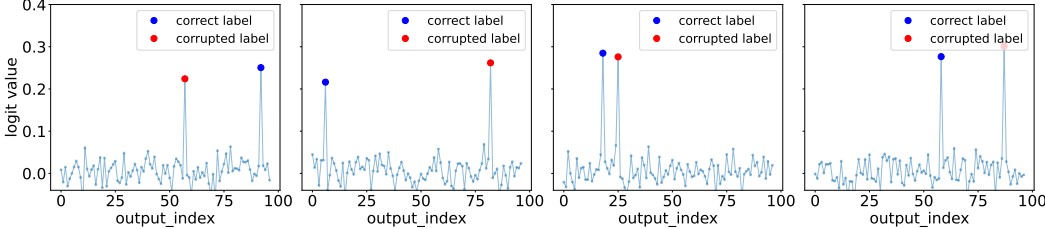

(b) *Partial Inversion* phase (wd=5.0). The logits corresponding to corrupted labels have similar values to the correct ones, which explains partial memorization of corrupted data in this phase.

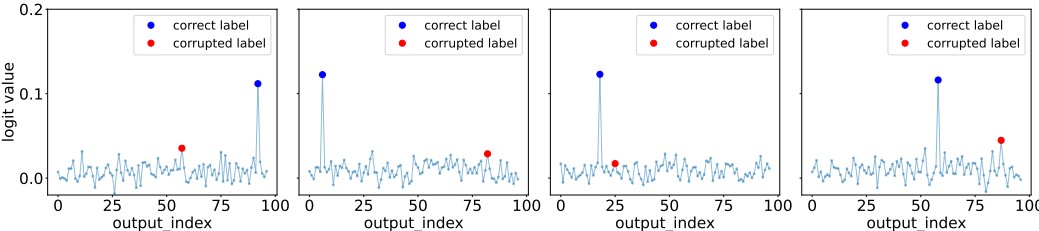

(c) *Full Inversion* phase (wd=15.0). The logits corresponding to corrupted labels are heavily suppressed compared to the correct ones, which explains the absence of memorization of corrupted data in this phase.

Figure 23: Network output logits on 4 different (corrupted) training examples examples, in various phases ($N = 500, \alpha = 0.5, \xi = 0.35$). Blue dots correspond to the logit value for the "correct" labels. Red dots correspond to the logit value for the corrupted labels.

## L    DETAILED PHASE DIAGRAMS

Here we replot the phase diagrams in Figure 3 together with train and test accuracies.

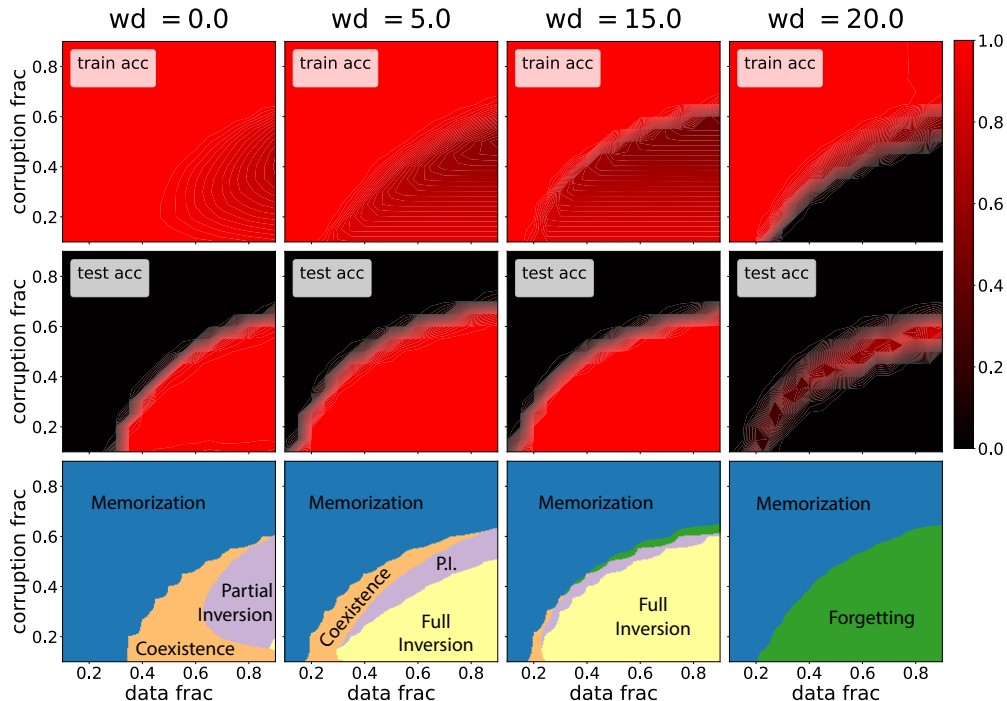

Figure 24: Modular Addition with weight decay. The first two rows are train and test accuracies.

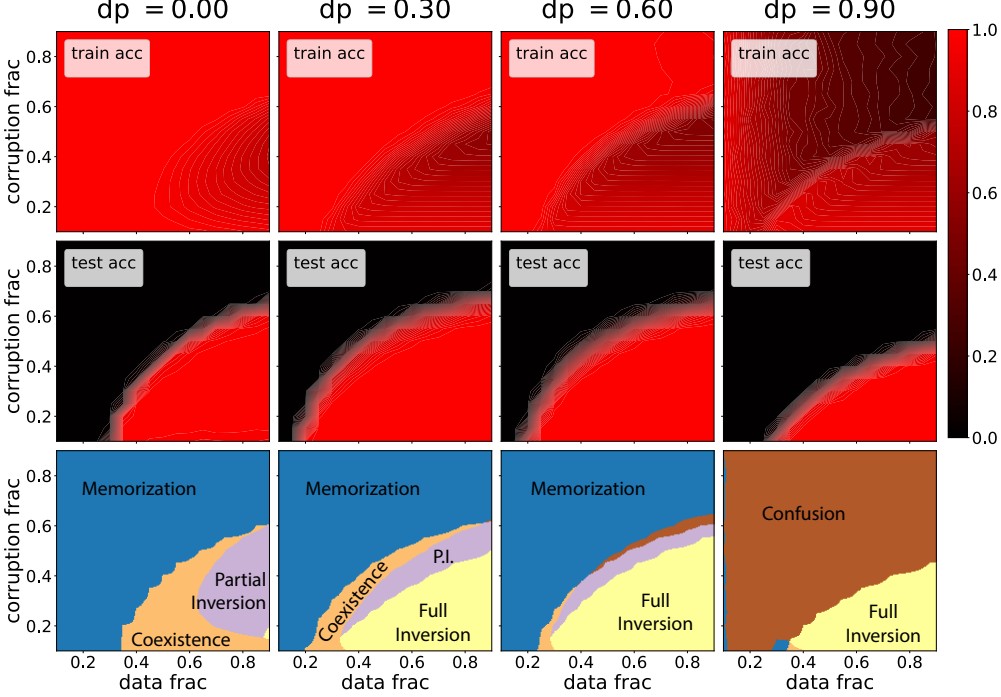

Figure 25: Modular Addition with Dropout. The first two rows are train and test accuracies.

## M    CHOICE OF LOSS FUNCTION

CrossEntropy loss is known to be sensitive to label noise (Ghosh et al., 2017). By adding regularization, we recover similar phase diagrams. The requirement for regularization is much greater for CrossEntropy loss compared to MSE. We find that CrossEntropy training is very sensitive to weight decay, with *Inversion* occuring for a narrow band of weight decay values, beyond which we find *Confusion*. We do not observe a *Forgetting* phase in this case. See Figure 26. With Dropout, we find that CrossEntropy training requires very high dropout probabilities to grok and/or attain *Inversion*.

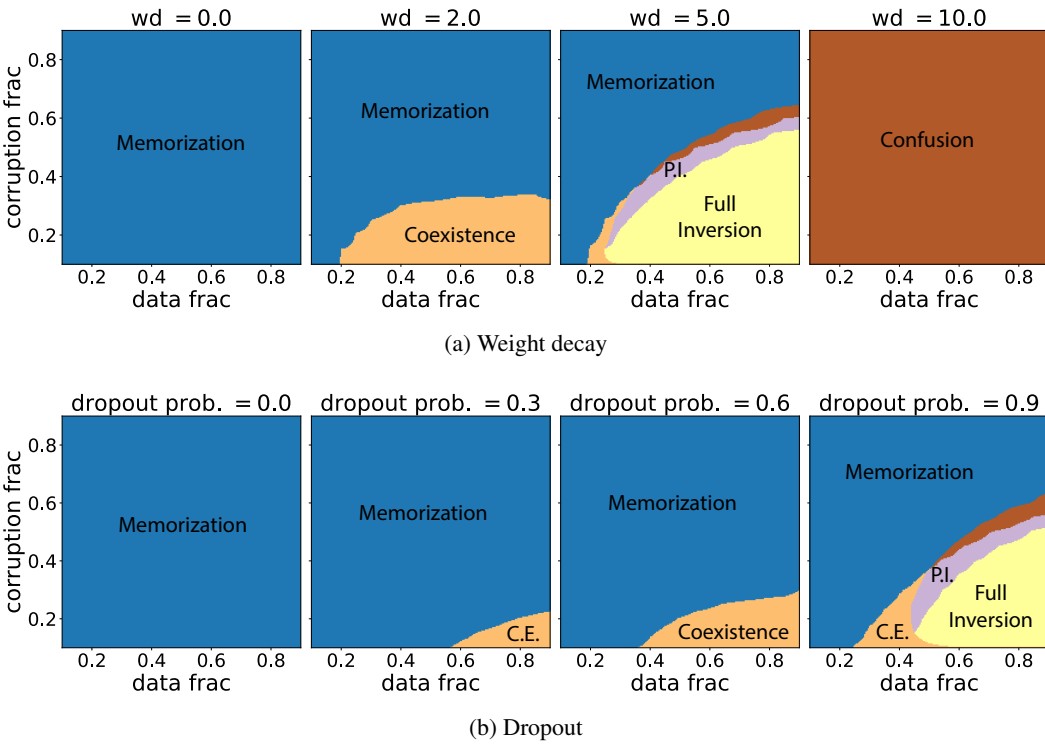

Figure 26: Modular Addition trained with CrossEntropy Loss. Robustness to label corruption is weaker compared to MSE.

## N    CHOICE OF ACTIVATION FUNCTION

To check if the story depends on the activation function, we did the modular addition experiments for 2-layer ReLU network, see Figures 27 and 28. The requirement for regularization for grokking is higher compared to quadratic activation function. We noticed that the phase diagram for weight decay behaves similarly to quadratic activation functions, except for the disappearance of the *Forgetting* phase. Dropout does not seem as effective for ReLU networks, with the network entering *Confusion* phase for even moderate dropout probabilites.

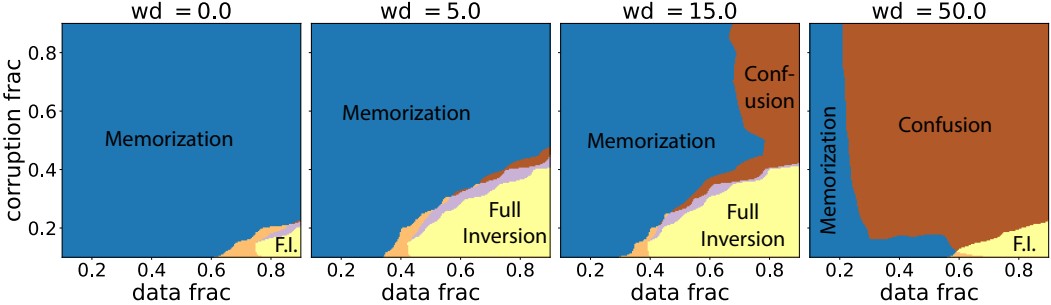

Figure 27: Phase diagram with weight decay, ReLU.

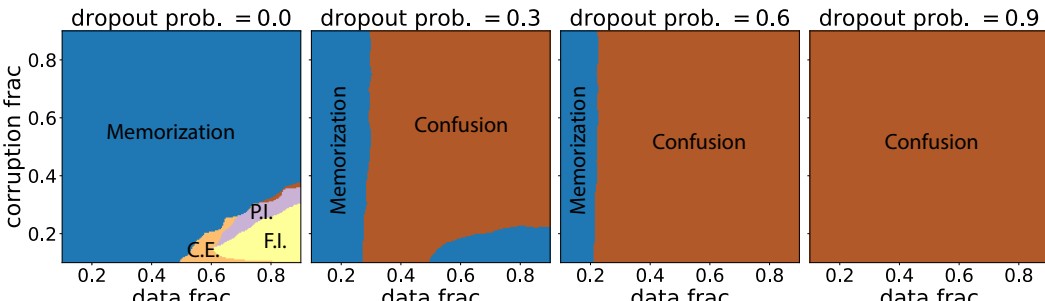

Figure 28: Phase diagram with Dropout, ReLU.

## O  EFFECT OF OF WIDTH

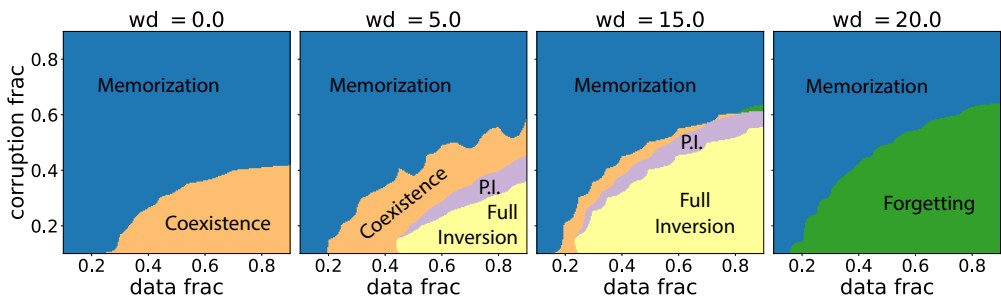

Figure 29: Phase diagram for width $N = 5000$ fully connected network with quadratic activation. Regularization with weight decay. Compared to $N = 500$ network in Figure 3(a), the network generalizes marginally poorly. Note: the absence of *Partial Inversion* phase and smaller *Coexistence* phase with $wd = 0.0$. This can be attributed to the increased capacity to memorize due to abundance of Neurons.

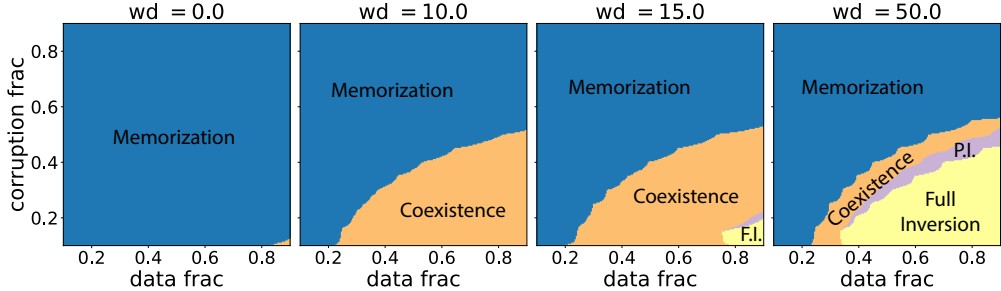

Figure 30: Phase diagram for width $N = 5000$ fully connected network with ReLU activation. Regularization with weight decay. In this case, the network performance is drastically improved compared to $N = 500$ (Figure 27). The confusion phase is largely eliminated, due to the increased capacity of the network, as a result of abundance of neurons.

## P  OTHER MODULAR ARITHMETIC DATASETS

In this section, we check if the phase diagram story is unique to modular addition. We repeat our experiments with modular multiplication ($z = (mn)\%p \equiv mn \mod p$) and obtain almost identical phase diagrams to modular addition (see Figure 31).

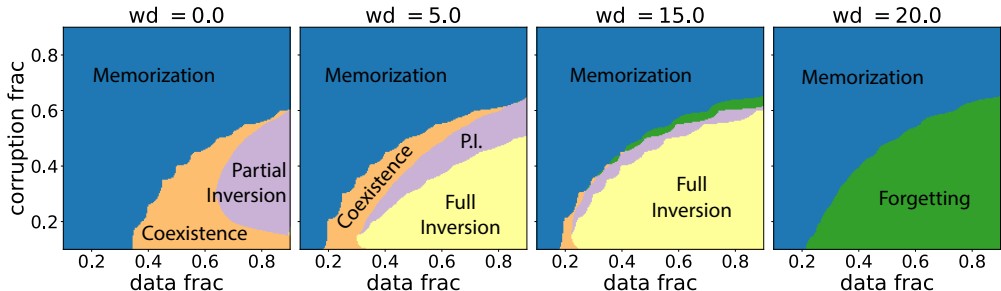

Figure 31: Phase diagram for Modular Multiplication, weight decay. All other settings are identical to Figure 3. Note that the phase diagram is identical to Modular Arithmetic case – representative of the similarity in the algorithms learnt by the network.

# Q HOW MANY NEURONS CAN BE PRUNED WHILE RETAINING GENERALIZATION?

We showed in the pruning experiments (Figures 6, 17) that a significant proportion of neurons (starting from low IPR) can be pruned out without decreasing test error. In Figure 32 we show how many neurons can pruned from a trained network at various widths. We train networks with widths between 500 and 10000; with learning rate 0.005 and weight decay 1.0, with AdamW optimizer and MSE loss. We use corrupted modular addition data, with data fraction $\alpha = 0.5$ and corruption fraction $\xi = 0.35$. We repeat the experiment 40 times, averaging over different random seeds.

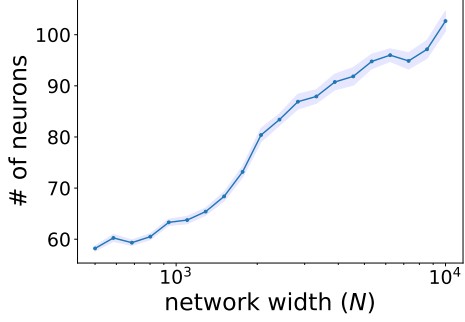
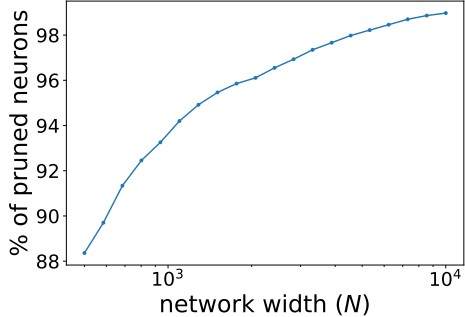

(a) Number of neurons required after pruning (from low IPR) to retain generalization. We see a sub-linear increase in the required neurons with width. shaded region denoted the standard error over 40 runs.

(b) % of neurons that can be pruned (from low IPR) while retaining generalization.

Figure 32: Pruning as many low IPR neurons as possible while retaining generalization, for various network widths ($\alpha = 0.5, \xi = 0.35$). The curves are averaged over 40 random seeds. Note that the x-axes on both plots have a logarithmic scale.

