# OpenReview forum: "To Grok or not to Grok: Disentangling Generalization and Memorization on Corrupted Algorithmic Datasets"
_ICLR.cc/2024/Conference — ICLR 2024 poster_

### Official Review · Reviewer_4cwR · 2023-10-18

**Soundness:** 3 good
**Presentation:** 3 good
**Contribution:** 2 fair
**Rating:** 5
**Confidence:** 4

**Summary:**

This paper studies the interplay between memorization and generalization in a controlled setting of synthetic task, where analytical solution exists and can be found by neural network optimizers. It demonstrates interesting behavior where generalization and memorization co-exists, and identified subnetworks that are responsible for each behavior. It further studies how popular regularization techniques impact the interplay between memorization and generalization.

**Strengths:**

This paper presented a simple synthetic machine learning setting where memorization and generalization can be studied in a very controlled manner. It further demonstrate that memorization and generalization could co-exist, and when it happens, there are clear distinction of subnetworks (at neuron level) that are responsible for each aspect. Moreover, by using the inverse participation ration (IPR), those neurons can be easily identified. This allows the authors to conduct further analysis on how different regularization techniques impact generalization and memorization from the perspective of memorization neurons. For example, it is shown that batch normalization operates in a very different manner from dropout and weight decay. Although the task of study is completely synthetic, it is still valuable to have such a controlled task that demonstrate interesting neural network behaviors such as grokking and memorization/generalization.

**Weaknesses:**

While some of the results are interesting, this paper is not strong in novelty and depth.

1. The main synthetic task studied in this paper is from previous papers, including the construction of the analytical solution and the matching neural network architecture that could realize such solutions. The main technique to distinguish memorization and generalization neurons, the inverse participation ration (IPR), is also from previous literature. While this paper does has some interesting observations when analyzing the memorization-generalization behaviors, I think it could benefit significantly from more novel analytical or algorithmic contributions.

2. The title and motivation put a lot of emphasis on grokking, yet there are no in-depth analysis of the grokking behaviors presented in the paper. I think the paper could be improved if it could show that the analytical tools used in this paper could bring new insights to our understanding of the grokking behaviors.

3. The experiments could be improved by more in-depth analysis. See below for a few examples.

    1. The paper shows regularizers help improve the generalization and demonstrated the "inversion" behaviors. However, given the emphasis on "grok" from the motivations, I would expect more in-depth analysis and insights to how those regularizers impact the *learning dynamics* that lead to the end result of "inversion" or "coexistence".

    2. In the case of co-existence, are there any consistent patterns on which neurons would become memorization and which would become generalization? Are they mostly in the lower layer or the upper layer? Do they have different learning dynamics?

    3. The paper studies the two-layer neural networks with quadratic activation function because analytical solution could be admitted with this architecture. However, most of the empirical studies (e.g. the phase diagrams) does not rely on this property. I think it is very helpful to see experiments with more realistic deep neural networks and diverse neural network architectures to see if they demonstrate consistent behaviors, and if not, to analyze what contribute to the different behaviors. There are some experiments in the appendix but only with small variations of the activation function or the width, and without in-depth analysis.

**Questions:**

1. The paper uses p=97. Why such a value is chosen? Does the behavior critically depend on the actual value of p (odd, even, prime, very small, very large, etc.)?

2. In Fig.6, the test accuracy remains almost perfect even after 60% of the neurons being pruned. How does the ratio (60%) depend on various aspects of the underlying task? Does it remain constant when the network size changes?

---

> ### Author Response · Authors · 2023-11-20
> **Response (1/2)**
>
> We thank the reviewer for the comprehensive feedback and detailed suggestions.
>
> ## Novelty
> Please refer to the corresponding section in the Global Response.
>
> ## Experiments with diverse architectures
> Please refer to the corresponding section in the Global Response.
>
> ## Impact of regularization on learning dynamics
> Please refer to the corresponding section in the Global Response.
>
> ## Choice of p
>
> Neural networks can generalize on modular addition tasks for all natural numbers $p$, given a large enough width. Furthermore, we find this to be true even on more general modular arithmetic tasks (such as $(mn)\\%p$ and $(m^2+n^2)\\%p$, etc.).
> However, formally, general modular arithmetic tasks are only well-defined over finite fields -- which require the modular base to be a prime number. We find that network training circumvents this subtlety in many cases.
>
> The size of the dataset also depends on this choice -- the number of total examples is $p^2$. Hence, a very small $p$ (such as $2,3,5$) would prevent learning. On the other hand, a very large $p$, while learnable, would demand excessive computing resources. The prime $p=97$ is the choice from the original paper on Grokking by Power et al. (2022). This choice strikes a reasonable balance between the aforementioned factors, but is in no way unique.
>
> ## How many neurons can be pruned without losing performance?
>
> We would first like to clarify that perfect generalization in the pruning experiment (Figure 5) remains **even beyond** pruning $60\\%$ of the low IPR neurons. Figure 6(left) showed only up to $70\\%$ pruning to explain the (sole) effect of pruning out low IPR neurons. We have added the extended version of this pruning experiment in Appendix I.1 (Figure 17(a)), where one can see that it is possible to prune $\\sim 88\\%$ of the neurons without losing test performance. We find that for $N=500$, the number of remaining neurons after maximal pruning is $\\sim 60$ ($N$ is the network width).
>
> According to the analytical solution (Equation 2), $\\lceil p/2 \\rceil$ neurons with different frequencies are required for the network to generalize correctly. Empirically, in real networks, the periodicity in neurons is approximate and the neuronal frequencies are not always unique. Consequently, real networks require more than $\\lceil p/2 \\rceil$ neurons for proper generalization. This is reflected in the aforementioned number $60$ being higher than $\\lceil 97/2 \\rceil=49$.
>
> We also empirically find that wider networks are more prone to such effects, and hence require more neurons for generalization. We have added this experiment in Appendix Q (Figure 32). As such, we see a monotonic, albeit sub-linear, increase in the number of neurons required with increasing network width. Note that the ratio of prune-able neurons *increases* substantially due to the sub-linearity with width. For N=10000, we find that $\\sim 99\\%$ of the neurons can be pruned while retaining generalization. In this case, the network requires $\sim 100$ neurons to maintain generalization.
>
> Moreover, the number of prune-able neurons also weakly depends on the phase in which the network lies (coexistence/partial-inversion/full-inversion.) Intuitively, due to the scarcity of periodic neurons in the coexistence phase, the network seems more economical, compared to the inversion phases. These comparisons are explained in Figure 17 of Appendix I.1 of the revised manuscript.
>
> ## New insights on grokking
>
> Grokking on modular arithmetic datasets has been observed in multitudes of setups with different architectures, optimizers, regularization schemes (or lack thereof), loss functions, etc. These setups, despite having very different dynamics, feature similar grokking behaviours. Consequently, we focus our attention on the common denominator, namely the dataset. We study how changing/corrupting the data affects these behaviors and also present new behaviors emerging out of this examination. As we mention in the Novelty section, on modular arithmetic datasets, label corruption doesn't just introduce conflicting examples -- it leads to a new modular arithmetic task. Thus, robust generalization on the target task with significant label corruption points to a bias towards a certain class of modular tasks, possibly ones that can be learnt with an algorithm similar to Hypotheses 2.1, 2.2. We believe that these insights are valuable for future investigations on grokking.
>
> Another important aspect of grokking is the tussle between memorization and generalization. With that in mind, we demonstrate that one can tune the generalization/memorization performance by tuning label corruption and regularization. We also characterize these behaviors using phase diagrams and interpret them using detailed IPR results. Moreover, our analysis also generalizes to various network architectures. All in all, we think that our experiments and tools will significantly advance the understanding of the subject.

---

> > ### Author Response · Authors · 2023-11-20
> > **Response (2/2)**
> >
> > ## Patterns of generalizing/memorizing neurons in the *coexistence* phase
> >
> > We can predict the occurrence of generalizing neurons before the grokking transition occurs by keeping track of per-neuron IPR during training. Furthermore, one can predict these behaviors during early training times by tracking the per-neuron IPR in the gradient updates -- the increase in IPR of gradients precedes that in weights.
> >
> > However, predicting such behaviors *at initialization* is difficult using IPR. As such, we do not find a consistent correlation between the Fourier spectrum of the weight vectors at initialization and their periodicity after training. We view the occurrence of periodic features and the partition of neurons into generalizing and memorizing neurons as emergent collective phenomena -- a comprehensive theory of grokking dynamics may give us further insight into the question.
> >
> > > Are they mostly in the lower layer or the upper layer?
> >
> > We are unsure about what the reviewer means by this question. We urge the reviewer to clarify their question so that we can give an accurate response.
> >
> > To avoid any possible confusion, we clarify the point about periodic neurons. In our 2-layer setup, *all* the weight matrices have periodic features. Specifically, using notation from the paper, the $k^{th}$ row of $U$, $k^{th}$ row of $V$ and $k^{th}$ column of $W$ matrices connect to a given $k^{th}$ (hidden) neuron. All three of these row/column vectors are found to be periodic with identical frequency. The per-(hidden)-neuron IPR histograms presented in the paper average the IPR over these three vectors. We refer the reviewer to Equation 4 and the surrounding text for further clarification. It can also be seen from the analytical solution that this has to be the case.

---

> > > ### Comment · Reviewer_4cwR · 2023-12-04
> > > **Response to the Authors**
> > >
> > > I would like to thank the authors for the detailed response and clarifications. The added results on more architectures and non-synthetic dataset (MNIST) also improves the paper. I still think some insights to help understanding the grokking behavior itself would be more exciting when the paper is mainly introducing another synthetic grokking task.

---

### Official Review · Reviewer_j4fN · 2023-10-26

**Soundness:** 3 good
**Presentation:** 4 excellent
**Contribution:** 3 good
**Rating:** 8
**Confidence:** 4

**Summary:**

Grokking is a recent empirical phenomenon where training achieves zero error long before neural network models generalize. People have identified this phenomenon to explore memorization vs. generalization and acquire mechanistic understanding of training dynamics of neural networks.

In this paper, the authors present randomization experiments where a fraction of labels are corrupted, i.e., replaced by random labels. A simple arithmetic task, namely modular addition, is used to train a small 2-MLP network. According to levels of memorization and generalization, the trained neural networks are categorized into several regimes: (1) memorization (2) coexistence (3) partial inversion (4) full inversion (5) forgetting. Each regime corresponds to an explanation of the inner mechanism.

**Strengths:**

Overall, I find this paper very clear and interesting. The authors focus on a recent phenomenon, and by experimenting under idealized data and networks, phase transitions are identified, each with meaningful interpretations.

Some of the key contributions include:
1. Randomization experiments reminiscent of [1], which helps understand generalization of neural nets in the presence of label noise.
2. A clear phase transition phenomenon and separates different data generating regimes into interpretable regions. This may be helpful for understanding the inner workings of neural networks and training dynamics.
3. Clear measurements that quantify the behavior of the trained network.
4. Some insights into regularization techniques such as batchNorm and weight decay.

[1] Chiyuan Zhang, Samy Bengio, Moritz Hardt, Benjamin Recht, and Oriol Vinyals. Understanding deep learning requires rethinking generalization. In International Conference on Learning Rep- resentations, 2017.

**Weaknesses:**

This paper follows a line of papers that study grokking, and provides a refined analysis of the phenomenon instead of proposing more general principles.

1. I feel that compared with the initial phenomenon identified in earlier papers, this paper is in a way less innovative.
2. The scope of the analysis is limited, as toy datasets and toy neural nets are studied. It is unclear whether these analyses can generalize to practical settings.
3. This paper does not contain a detailed study of the training dynamics, how the weights behave, or some theoretical analyses. It is unclear whether this paper has impact on the theory community.

**Questions:**

See the above sections.

In addition, is there any scenario where we believe Partial Inversion or Full Inversion may be observed in practice?

---

> ### Author Response · Authors · 2023-11-20
>
> We thank the reviewer for the encouraging feedback.
>
> ## Novelty
> Please refer to the corresponding section in the Global Response.
>
> ## General settings
> Please refer to the corresponding section in the Global Response.
>
> ## Training dynamics
> Please refer to the corresponding section in the Global Response.
>
> ## Answers to Questions
>
> - Yes, we do observe partial as well as full inversion on Modular Arithmetic tasks learned with the Transformer architecture as well as deeper MLP networks (Section 2.2  and Appendix G). Even in these general settings, we can relate periodic weights and IPR to generalization.
> We also observe inversion in corrupted image tasks such as MNIST (Appendix H). For further details please refer to the Global Response.

---

### Official Review · Reviewer_kDnz · 2023-10-31

**Soundness:** 3 good
**Presentation:** 3 good
**Contribution:** 2 fair
**Rating:** 5
**Confidence:** 4

**Summary:**

This paper investigates the grokking phenomenon on algorithmic datasets with label corruptions.  In particular, a two-layer MLP is fitted to a modular arithmetic dataset with varying degrees of label corruptions, weight decay, and dropout rates. As a measure of generalization, the authors propose to use the inverse participation ratio, which quantifies the periodicity in the parameters. Experiments show that (i) the model can perfectly memorize the mislabeled examples while achieving near-perfect test accuracy, (ii) the model learns to "correct" some or all mislabeled examples depending on the level of regularization, (iii) the memorizing/generalizing neurons are identifiable.

**Strengths:**

The paper is written clearly and the results are discussed well. The specific idea of training under label corruption is interesting, which seems to allow for studying memorization on the neuron level. I find some results important (e.g. simultaneous memorization and generalization and sub-network identification) while some are not as significant (e.g. the impact of weight decay).

**Weaknesses:**

Recently, there have been a series of papers on grokking, most of which rely on empirical analysis of training dynamics or final networks. Although such findings could be thought-provoking, they often come with the risk that findings/questions may apply to the particular setting considered in the work. I'm not sure whether the analysis and findings in this work can be extrapolated to other setups, e.g., different architectures or datasets. This is why I'm closer to a rejection but would be happy to discuss it with the authors and other reviewers.

- The paper analyzes the learning under label corruptions in the same vein as previous works. As shown in Nanda et al. (2023), the algorithm implemented by the algorithm after the grokking phase, roughly speaking, boils down to a handful of periodic functions, which is further confirmed by Gromov (2023) in terms of an analytical solution. Based on this observation, the authors propose to use IPR as a measure of "how periodic a neural representation is", which serves as a proxy for whether a neuron generalizes or memorizes. Then, the effect of pruning individual neurons is investigated in the same spirit as Nanda et al. (2023). All in all, on the procedural side, the only novelties seem to be the use of IPR and label corruption.

- I find the finding that there are separate generalizing/memorizing sub-networks (or rather collections of neurons) interesting. Yet, the identification of these sub-networks is demonstrated on this toy algorithmic problem whose solution is known to be periodic. Consequently, I'm not sure if this finding translates into more general problems with non-periodic subnetworks that generalize.

- I do not understand the main takeaway of sections 3.1 and 3.2. Is there any new finding or just a confirmation of well-known results?

**Questions:**

- Interestingly, even in the case of full inversion without batch normalization, low IPR neurons do not disappear. Why does this occur?
- Likewise, batch normalization seems to downweight the low IPR activations but does not zero them out, meaning that memorization still occurs. How do the authors interpret this?
- Multi-modality in Figure 4 alone does not imply low/high IPR neurons memorizing/generalizing as there is no evidence that low IPR neurons are indeed connected to memorization. I think this becomes clear only in the second to the last paragraph of section 2.
- _In trained networks, a larger population of high-IPR neurons leads to better generalization_ requires citation or explanation.
- _Choosing quadratic activation function ... makes the problem analytically solvable._ requires citation.
- _It’s role ..._ <--- typo
- _In the previous Section_ <--- S should be small

---

> ### Author Response · Authors · 2023-11-20
>
> We thank the reviewer for their detailed feedback and suggestions.
>
> ## General architectures and datasets
> Please refer to the corresponding section in the Global Response.
>
> ## Novelty
> Please refer to the corresponding section in the Global Response.
>
> ## Section 3
>
> In the previous version of the manuscript, we dedicated Section 3 to summarize the effect of various regularization techniques and compare these effects with existing literature.
> In the revised version, considering the reviewers' suggestions, we have shortened this discussion in favor of the results for general architectures -- Transformers and deeper MLPs.
>
> ## Low IPR neurons in full inversion phase
>
> We encode labels as $p$-dimensional one-hot vectors, so the accuracies depend only on the largest output logit. On the other hand, the optimization objective depends on the MSE loss over \emph{all} the logits. In the presence of label corruption, the MSE loss naturally includes the corrupted training data. For these corrupted training examples, we find that the output logit-vectors contain two competing peaks -- corresponding to the corrupted and the "correct" labels (Appendix K, Figure 23). This underlies the tussle between generalizing and memorizing network predictions. Regularization techniques such as weight decay suppress the corrupted-logits compared to correct-logits. This results in the *full inversion* phase we observe. The few remaining low-IPR neurons still contribute to a weak peak and lower the training loss, but they do not affect the network predictions or accuracy. The effect of these low IPR neurons on loss can be seen in the pruning curves in Appendix I.2 (Figure 18(c)).
>
> ## Effect of BatchNorm
>
> Indeed, BatchNorm does not zero-out the low IPR neurons, but merely downweights them. As a result, the regularization effect of BatchNorm is less substantial compared to weight decay and dropout. This can be seen by comparing the phase diagrams in the Figures 4, 8. We emphasize that the regularizing effect of BatchNorm, albeit weaker, is noteworthy due to its interpretability.
>
> Similar to the logits in Figure 23, we also see multiple peaks in the presence of BatchNorm. The suppression of low IPR neurons with BatchNorm suppresses the peak corresponding to the corrupted label in favor of the correct one, which explains the regularizing effect of BatchNorm.
>
> ## Corrections
>
> We thank the reviewer for pointing out the typos and additional citations. We have made these corrections in the revised manuscript.

---

> > ### Comment · Reviewer_kDnz · 2023-11-22
> > **my response to author response**
> >
> > Thanks for the author's response. I'm sorry if I'm being "the annoying reviewer 2" but I really am having difficulty seeing the main gist of the paper. I understand that this work introduces a way of analyzing neurons to identify where generalization/memorization happens. Although these findings are interesting as such, I don't see how they would translate into virtually any other machine learning problem. The first grokking paper presented a shocking finding, which was then (attempted to be) replicated in various setups. Yet, this work examines an even more constrained setup (with label corruptions). As such, it does not take "generalization vs memorization" in a broad sense. More concretely, I don't see how "these ideas allow for a quantitative distinction between generalizing and memorizing sub-networks, deepening our understanding thereof in trained neural networks." What knowledge can I transfer from this paper to "memorization in ResNets trained on ImageNet"?
> >
> > Second, I disagree with the claim that "the resilience of modular arithmetic datasets to label corruption is also a non-trivial and significant finding". What makes this finding important? How does it guide future grokking research? Further, the "bias towards "simpler" polynomials introduced by regularization techniques" was already shown in the "mechanistic interpretability" paper. Even more importantly, Figure 6c of "Towards Understanding Grokking: An Effective Theory of Representation Learning" paper already shows grokking happens without regularization.
> >
> > These being said, I promise the discuss the significance of the contributions with the other reviewers and revise my score if I'm convinced.

---

> > > ### Author Response · Authors · 2023-11-23
> > > **(Resopnse)^3**
> > >
> > > We thank the reviewer for their prompt response and follow-up comments. We welcome all the constructive feedback and are eager to communicate/clarify our viewpoint.
> > >
> > > ## General settings
> > >
> > > For clarity, we re-emphasize the two different avenues of generalizing our results; namely, (i) general architectures, and (ii) general datasets.
> > >
> > > On the general architecture side, as we mentioned in the General Response, we have shown that our results extend to Transformers and deeper MLPs (new Section 2.2 and Appendix G in the revised manuscript).
> > >
> > > On the general dataset front, we agree that analyses using IPR cannot be carried over verbatim to general datasets like ImageNet. However, we do see that image datasets like MNIST have similar phases and inversion phenomena as modular arithmetic datasets (Appendix H). Thus, they feature the same phenomena, albeit with a different (unknown) algorithmic bias. The challenge for future research is to identify this bias, which would allow a way to find these sub-networks for general datasets. To that end, we believe that our methodology, i.e., introducing incremental label corruption to the dataset and then finding the invariant features in various phases, could be a useful method for understanding memorization and generalization in a broader setting.
> > >
> > > ## Resilience of modular arithmetic tasks to label corruption
> > >
> > > We first clarify that the word "simpler" means completely different things in the paper by Nanda et al. compared to ours.
> > >
> > > - In their work (Appendix E.1), they speculated a vague notion of simplicity: "memorization complexity scales with the size of the training set, whereas generalization complexity is constant. The two must cross at some point!"
> > > They conjectured that there are only two solutions, namely, a memorizing solution (complex) and a generalizing solution (simpler). Both correspond to the *same* task, and regularization would bias the network towards the simpler one.
> > >
> > > - Whereas the point we are trying to convey is that, given some training data, there are an exponentially large number of solutions and they all correspond to *some* modular arithmetic polynomial. Then, simplicity refers to the polynomial that the network learns upon training.
> > >
> > > To clarify this idea, let's view the learning modular arithmetic task as "completing the $p\times p$ table"; where the entries of the table correspond to the map from $(m,n)\rightarrow z$. The network has to complete the table given a fraction of the entries, a.k.a. the training data.
> > >
> > > First, let's look at the uncorrupted training set. In this setup, every possible completion of the table corresponds to a resulting polynomial, the "target polynomial" being one of them. The fact that the network can generalize, already shows a preference for the simple target polynomial compared to other completions. This is known from the very first paper on grokking; even before the work by Nanda et al. Moreover, this bias remains even in the absence of any regularization.
> > >
> > > Whereas in our case, a significant portion (as high as $50\\%$) of the training data (i.e. pre-filled table) has corrupted labels. Since *any* table corresponds to a polynomial in modular arithmetic tasks, the model could have recognized the data as some uncorrupted training data from another polynomial. The fact that regularization can help the network overcome the penalty of making "wrong" predictions on the training set (*inversion* phase) and finding the correct simpler polynomial is highly non-trivial. This inductive bias is different and much stronger than the uncorrupted case.
> > >
> > > To summarize, our findings on corrupted datasets points to an alternative viewpoint to the inductive bias on modular arithmetic tasks. We believe that future research should focus on explaining the (strong) bias towards a certain small set of modular polynomials amongst the astronomically large set. This may prove crucial in explaining why certain modular polynomials are un-grokkable; for example $(m^3 + mn^2 + n) \\% p$ [Power et al. (2022)].

---

### Author Response · Authors · 2023-11-20
**Global Response (1/2)**

We thank all the reviewers for their constructive feedback and useful suggestions. Since the question about novelty, the applicability of our work to general settings and training dynamics was raised by multiple reviewers, we address them in this Global response.

## General architectures and datasets
We have added results for modular arithmetic tasks learned with the Transformer architecture as well as deeper MLP networks in Section 2.2 and Appendix G of the revised manuscript. We find phase diagrams with coexistence and inversion phases upon adding label noise. We find that in these general settings, the input/embedding and (in some cases) the output layers have periodic weights, which can be correlated with generalization using IPR. The remainder (bulk) of the network serves to combine the periodic features nonlinearly. We also find that weight decay is more effective than Dropout and BatchNorm in preventing the memorization of corrupted labels for these general architectures.

Additionally, we also observe coexistence and inversion phases on corrupted image datasets such as MNIST (Appendix H). While the quantification of generalizing/memorizing features depends on the interpretability of the modular arithmetic tasks, the effects of label noise and regularization appear in more general problems. We hope to pave the way for analyses similar to ours in these diverse cases.

## Novelty

With regards to the novelty of our work, we would like to emphasize the following:

1. From our viewpoint, the two important characteristics of grokking behaviors are the properties of the dataset and the tussle between generalization/memorization. We engineer our setup to address both, by combining label corruption with the interpretable modular arithmetic task. We disentangle generalization and memorization in the presence of label corruption on this dataset; and characterize various training phases. We also pinpoint the effect of label corruption by using distributions of per-neuron IPR and IPR-based pruning. Neither of these analyses has been performed in prior works. These ideas allow for a quantitative distinction between generalizing and memorizing sub-networks, deepening our understanding thereof in trained neural networks. Furthermore, our work sheds light on the effect of regularization on trained networks, a topic not thoroughly examined in previous studies. We quantify the effect of various regularization schemes like weight decay, dropout and BatchNorm; and provide a detailed comparison -- which is particularly valuable. We would also like to point out that regularization using Dropout or BatchNorm has not been previously studied on this task.

2. The resilience of modular arithmetic datasets to label corruption is also a non-trivial and significant finding. These tasks are defined over a finite field, where each map from $(m,n)\rightarrow z$ (where $m,n,z \in [p]$) corresponds to a distinct polynomial over the field. Thus, each instance of label corruption gives rise to a (unique) alternative polynomial (i.e. a new modular arithmetic task), different from the target polynomial. This is in contrast to image datasets, where label corruption leads to conflicting examples. The network's ability to discern the intended task with such a corruption is noteworthy. It shows a bias towards "simpler" polynomials introduced by regularization techniques.
We surmise a bias of neural networks towards learning modular arithmetic tasks that can be learnt using the algorithm mentioned in Hypotheses 2.1, 2.2. (Note that such an algorithm applies to more general tasks than addition.)

3. The correlation between generalization and IPR extends beyond the 2-layer MLP setup, to general architectures. We empirically demonstrate this in the added results to the updated version (Section 2.2 and Appendix G). This finding suggests a universal framework for understanding generalization on modular arithmetic datasets. We also show inversion behaviors on MNIST (Appendix H, Figure 16); and posit that for any given dataset, comprehending the model's learned features through the lens of label corruption is feasible. This is particularly relevant for non-synthetic datasets where label corruption is a common occurrence.

---

> ### Author Response · Authors · 2023-11-20
> **Global Response (2/2)**
>
> ## Training Dynamics
>
> Grokking dynamics and the impact of regularization is an unsolved problem even in the absence of label corruption. Adding label corruption introduces additional behaviours to this phenomenon -- for instance, we find a second transition after grokking in the full inversion phase, where the network unlearns the memorizing representations. Studying the dynamics of grokking as well as these new behaviours is extremely relevant and interesting, but it lies beyond the scope of our work. The goal of our work is to utilize the tools such as IPR and label corruption to disentangle generalization and memorization in trained networks, as well as interpret the effect of various regularization techniques on these networks. To that end, we perform detailed experiments including phase characterizations, IPR distributions and pruning. The exact dynamics that lead to such training behaviours requires a separate line of careful investigation, which we leave for future work.

---

### Meta-Review · Area_Chair_mQE2 · 2023-12-05

**Metareview:**

This paper considers the robust generalization problem for modular arithmetic tasks. The main results show that it is possible for a network to memorize the corrupted label and achieve generalization, and when there is regularization can ignore the corrupted data. The reviewers find some of the results interesting, although there are also concerns about the novelty of the paper.

**Justification For Why Not Higher Score:**

The paper is already borderline accept/reject.

**Justification For Why Not Lower Score:**

While I do share some of the concerns for novelty, the label noise setting was not considered before and was interesting to me (it's true that label noise was widely considered elsewhere, but not in this modular arithmetic task). I'm OK either way.

---

### Decision · Program_Chairs · 2024-01-16

Accept (poster)